# Prevalence of unculturable bacteria in the periapical abscess: A systematic review and meta-analysis

**Alaa Muayad Altaie**[1,2], **Basema Saddik**[1,3], **Mohammed Amjed Alsaegh**[1,2], **Sameh S. M. Soliman**[1,4], **Rifat Hamoudi**[1,5,6‡]*, **Lakshman P. Samaranayake**[1,2‡]*

1 Research Institute for Medical and Health Sciences, University of Sharjah, Sharjah, UAE, 2 Department of Oral and Craniofacial Health Sciences, College of Dental Medicine, University of Sharjah, Sharjah, UAE, 3 Department of Family and Community Medicine, College of Medicine, University of Sharjah, Sharjah, UAE, 4 Department of Medicinal Chemistry, College of Pharmacy, University of Sharjah, Sharjah, UAE, 5 Department of Clinical Sciences, College of Medicine, University of Sharjah, Sharjah, UAE, 6 Division of Surgery and Interventional Science, University College London, London, United Kingdom

‡ These authors are joint senior authors on this work.
* rhamoudi@sharjah.ac.ae (RH); lakshman@hku.hk (LPS)

## Abstract

### Objective

To assess the prevalence of unculturable bacteria in periapical abscess, radicular cyst, and periapical granuloma.

### Methods

PubMed, Scopus, Science Direct, and Ovid databases were systematically searched from January 1990 to May 2020. All the included studies were cross-sectional design. The risk of bias was assessed using Joanna Briggs Institute check-list. Heterogeneity was described using meta-regression and mixed-effects model for lesion, country, and sequence technique moderators. Funnel plot and unweighted Egger's regression test were used to estimate the publication bias. Microbiome data on diversity, abundance, and frequency of unculturable bacteria in the periapical lesions were reviewed, analysed, and the principal component analysis (PCA) was performed.

### Results

A total of 13 studies out of 14,780, were selected for the final analysis. These studies focused on the prevalence of unculturable bacteria in periapical abscesses and related lesions. Approximately 13% (95% CI: 7–23%) of the cumulative number of bacteria derived from periapical abscesses was unculturable. Country moderator significantly (P = 0.05) affects the diversity summary proportion. While the pooled frequency of unculturable bacteria was 8%; 95% CI: 5, 14%, the estimate of the pooled abundance of unculturable bacteria was 5%; 95% CI: 2, 12% with a significant (P = 0.05) country moderator that affects the abundance summary proportion. Of the 62 unculturable bacteria, 35 were subjected to PCA and *Peptostreptococcus* sp. oral clone CK035 was the most abundant species in periapical

**Funding:** The authors received no specific funding for this work.

**Competing interests:** The authors have declared that no competing interests exist.

abscesses. Hybridization techniques were found to be the most reliable molecular methods in detecting the abundance and frequency of unculturable bacteria.

## Conclusion

The significant prevalence of unculturable bacteria in the periapical abscess, suggests that they are likely to play, a yet unknown, critical role in the pathogenesis and progression of the disease. Further research remains to be done to confirm their specific contributions in the virulence and disease progression.

## Introduction

Periapical abscesses are, by far, the most frequent infectious lesions of the alveolar bones [1, 2]. Periapical abscess occurs in and around the apex of a root, the periodontal membrane of a tooth, and the adjacent alveolar bone [2–4]. The spread of infection via the apical foramen and inflammation sets in a cascade of reactions attracting inflammatory chemical mediators to initiate the periapical pathology that eventually results in a periapical abscess, a radicular cyst, or a periapical granuloma [2, 5].

There are several classifications of periapical lesions [6, 7]. World Health Organization (WHO) [6] provides a classification of the periapical lesions including periapical abscess, radicular cyst, and periapical granuloma, according to the clinical signs, other than symptoms or the histopathological differences of the periapical lesions. Another classification was proposed by Nair [7] and depending on the histopathological findings of the periapical abscess, radicular cyst, and periapical granuloma. However, it was the best to combine the clinical findings of signs and symptoms in addition to the histopathological examination when classifying the periapical lesions. This was achieved by using the updated version of classification of the American Association of Endodontics (AAE) for periapical lesions [8]. Periapical abscesses were classified into acute and chronic periapical abscess according to onset type, pain intensity, swelling, and discharge of pus or sinus formation [8]. The histological identification of periapical granuloma differentiates it from radicular cyst in which the former characterized by chronic inflammatory cells such as macrophages, plasma cells and lymphocytes and sometimes a cluster of multinucleated giant cells, capillaries, fibroblasts, and collagen fibres are also present [7, 8].

In clinico-pathological terms, periapical abscess is defined as a localized collection of pus within the alveolar bone, at the root apex of a tooth [9]. As mentioned, the lesion is usually initiated as a sequel of extension of infection into the periapex of an affected tooth. Once the intact pulp chamber is breached, colonization of the root canal ensues with a diverse mix of bacteria [10]. After entering the periapical tissues, these bacteria induce an acute inflammatory reaction and pus formation leading to a periapical abscess [11]. A number of researchers, over the last few decades, have investigated the microbiome of the periapical abscess using conventional bacteriological culture and noted that *Bacteroides* sp., *Streptococcus* sp., anaerobic cocci, and *Fusobacterium* sp. are the major constituents of periapical abscess [12]. However, recent next generation sequencing (NGS) studies indicate that a profusion of unculturable bacteria are present in periapical abscess and are the likely co-contributors to the disease [13].

Using polymerase chain reaction (PCR) and cloning strategies that target 16S rRNA, it is possible to determine the bacterial composition and diversity of any given infection [14]. The 16S rRNA approach defines a species (or more precisely a phylotype) as strains or clones

with > 98.5% identity [15]. Studies based on 16S rRNA gene clone libraries have shown that 40–60% of the oral microbiome is composed of as-yet-unculturable bacteria [16–18]. For instance, the Human Oral Microbiome Database (HOMD) lists approximately 220 oral taxa that have not been cultivated [19]. Thus, only 29–50% of the oral species-level taxa belonging to the phyla Firmicutes, Proteobacteria, Bacteroidetes, Actinobacteria, and Fusobacteria have been successfully cultivated. The number of cultivable members of the Spirochaetes and Synergistetes phyla is relatively low [16, 20]. Of 48 oral Spirochaetes listed in HOMD, only ten have been cultivated and named, while only three of oral Synergistetes, have been cultivated and named: *Jonquetella anthropi* [21], *Fretibacterium fastidiosum* [22], and *Pyramidobacter piscolens* [23]. Furthermore, it has been estimated that approximately 50% of the human oral flora is unculturable [24, 25], and the failure of curing of some infectious diseases speculated that at least some of the unculturable microbiome are involved in disease progression and may account for unknown antimicrobial resistance [26]. On the other hand, the importance of some unculturable bacteria represented by their production of secondary metabolites that may have a promising future as effective antimicrobial agents [27]. These findings indicate that a significant proportion of bacterial species inhabiting the oral ecosystem and causing dental diseases is yet to be described, and their contribution to diseases such periapical abscess is yet to be defined.

There are no reviews in the English Language literature, to our knowledge, on the unculturable bacteria in periapical abscess and other sequelae of pulp necrosis such as radicular cysts, and periapical granulomas. Hence, the primary aim of this study was to perform a systematic literature review and meta-analysis of the unculturable bacteria in periapical lesions. A secondary aim was to review the role of individual unculturable bacteria in terms of abundance and frequency in the foregoing periapical lesions, and to review the molecular techniques evaluating the latter. Therefore, the purpose of this study was to conduct a systematic review and meta-analysis that investigates the prevalence of unculturable bacteria in the periapical abscess, radicular cyst, and periapical granuloma using culture-independent methods. The followed question format was (CoCoPop) [28, 29] as condition, context, and population. The condition represents the unculturable bacterial profile, context represents all studies conducted on periapical abscesses, radicular cysts, and periapical granulomas using culture-independent methods, and population represents human clinical samples derived from periapical lesions (periapical abscess, radicular cyst, and periapical granuloma).

## Methods

### Protocol and registration

We performed the systematic review and meta-analysis in accordance with Cochrane Collaboration and Preferred Reporting Items for Systematic Reviews and Meta-Analyses (PRISMA) guidelines [30]. The study was registered in PROSPERO database https://www.crd.york.ac.uk/PROSPERO/ with the registration ID (CRD42020160557).

### Search strategy and study selection

First, we systematically searched PubMed, Scopus, Science Direct, and Ovid databases using the MeSH (Medical Subject Headings) terms "dentoalveolar abscess", "periapical lesion", "periapical abscess", "periapical granuloma", "radicular cyst", "unculturable bacteria", and "sequencing" (S1 Table). Manual searching, and cross-reference checks were also conducted. The search was limited to human studies published over a decade, from 1990/01/01 to 2020/05/31. The inclusion criteria for the study were, English language, and original studies on microbiological investigations using culture-independent methods. The exclusion criteria

were represented by any paper that used culture-dependent methods for bacterial identification, did not identify the unculturable bacteria, and histologically did not specify radicular cyst from periapical granuloma. Two independent reviewers (A.A.M, A.M.A) identified relevant studies based on the inclusion criteria and MeSH terms. Cohen's kappa was used to assess the inter-rater reliability between the two reviewers (~ 20 for title and abstract screening, and ~10 for full text screening) during the selection of studies for inclusion. Cohen's kappa less than 0.20 referred as slight agreement, between 0.21–0.40 as fair agreement, between 0.41–0.60 as moderate agreement, between 0.61–0.80 as substantial agreement, and between 0.81–1.00 as perfect agreement. Discrepancies between the reviewers were resolved after an iterative consensus process.

## Data extraction

Full-text articles were reviewed in detail after a thorough screening of the titles and abstracts. For each selected article, the general characteristics which included the first author's name and year, study design, country, sample size, gender, age, dentition type, numbers and lesion type, sample collection procedure and origin, and the molecular biology method for bacterial detection, were extracted to an Excel spreadsheet. When studies included more than one type of lesion, the corresponding information was recorded. Additional information including diversity, abundance, and frequency as well as the abundance and frequency of the individual organism were also reported.

## Risk of bias assessment

Risk of bias assessment was performed using a customised checklist based on Critical Appraisal of Joanna Briggs Institute for Analytical Cross Sectional Studies [31] as per the selected study criteria. This was used to assess the risk of bias, or the quality of studies included. The standard Checklist for Analytical Cross Sectional Studies was used, and each article independently assessed by the two authors (A.A.M, and A.M.A) (S2 Table). Cohen's kappa was used to check the inter-rater reliability for scoring the risk of bias assessment of the included studies. The final scores of the risk of bias assessment for the included studies were determined after consensus process between the reviewers.

## Resources used to identify the unculturability of bacteria

The following web resources were utilized to ascertain whether the identified bacteria are culturable or not. Expanded Human Oral Microbiome Database (eHOMD) [32] http://www.homd.org/, NCBI Taxonomy Browser-NIH [33] https://www.ncbi.nlm.nih.gov/Taxonomy/taxonomyhome.html/, and the Culture Collection of the University of Gothenburg [34] https://ccug.se/ were used to check the culturability of the bacteria included in the study.

## Statistical analysis

Meta-analysis was performed as primary outcomes for the diversity, abundance, and frequency of unculturable bacteria in the periapical abscess, using RStudio software version 1.2.5019 available at https://www.r-project.org/. Forest plot was conducted to graphically represent the consistency and reliability of the results. The analysis was done for three separate groups on the proportion (percentage) of unculturable bacteria [35]. Moderators were identified and included the lesion type, country of the study, and the molecular biological technique used for bacterial identification. The proportion in each group was set to less than 0.2 but as the study

number was relatively low, data transformation was performed using logit transformation to obtain the approximated, binomial distribution [36, 37].

Transformed proportions were combined to get the pooled effect sizes. Random-effect model [38] with DerSimonian and Laird method were used for this purpose [39]. The level of heterogeneity was assessed by Cochran's Q test and quantified using Higgins $I^2$ test [40] Heterogeneity was classified as low if the $I^2$ was (0–40%), moderate (30–60%), substantial (50–90%), or high (75–100%) [41, 42]. P-values were obtained by comparing the Chi-squared test with a statistic Q and k-1 degrees of freedom, using the confidence interval 95% as the cut-off for statistically significant of heterogeneity. The proportion of true heterogeneity across all studies was described using meta-regression of moderators and mixed-effects model with DerSimonian and Laird method to find the effect of these moderators on summary effect size.

Outlying studies were tested using studentized residuals that showed z-value of more than 2 or 3 in an absolute value. The decision of removing outlying studies was dependent on testing its influence on summary effect size [43]. Studentized residuals test has been used to estimate the change after removing the outliers [43] and to detect the normal distribution of studies [44]. To confirm the impact of the outlying studies, the following tests were employed to show the change in their corresponding estimate after removing the outliers. DFFITS and Cook's distances values showed the change in the standard deviations [44], co-variance ratios (cov.r) showed the effects on the precision of estimates, and the amount of heterogeneity (tau$^2$.del) displayed tau$^2$ of each study [44]. The test statistics for heterogeneity (QE.del) showed the $\chi^2$ test for homogeneity [44]. Hat values indicated that large values reveal large influence, and the weights test indicated the weight of a study in the overall meta-analytic average effect size. Lastly, funnel plot asymmetry and unweighted Egger's regression test were used to estimate the publication bias [35].

Secondary outcomes were performed for individual unculturable bacterial abundance and frequency using principal component analysis (PCA) in RStudio software. Factoextra, magrittr, and dplyr libraries were used to determine the individual unculturable bacteria in relation to their abundance, frequency, and biological sequencing techniques [45].

## Results

### Study selection

A literature search of PubMed, Scopus, Science Direct, Ovid, as well as a manual search retrieved a total of 14,780 studies on periapical abscess, radicular cyst, and periapical granuloma. Deduplicating reveals the removal of 39 studies. Screening the titles and the abstracts of these studies leads to the exclusion of 11,220, and 3343, respectively. The remaining 217 full-text articles were perused and 201 were excluded (Fig 1). The excluded studies were unspecified lesions (absence of histological analysis for discrimination between periapical granuloma and cyst) (60 articles), detection of culturable bacteria through molecular methods (45 articles), case-reports (22 articles), samples unrelated to granuloma, cyst, and abscess (16 articles), culture methods for bacterial detection (10 articles), histobacteriological detection of bacteria (10 articles), narrative reviews (eight articles), detection of bacteria through culture media then molecular methods, mass spectrometry, or electron microscope (six articles), unculturable bacteria had not been recorded by molecular methods (five articles), case-series (three articles), full-text articles are unavailable (three articles), detection of virulence genes (three articles), no bacteriological detection (three articles), nondescript unculturable bacteria (two articles), immunological detection of culturable bacteria (two articles), full-text article in language other than English (one article), book chapter (one article), and duplicate publication (one article). Of the final records, 16 were included in the qualitative analysis of the systematic

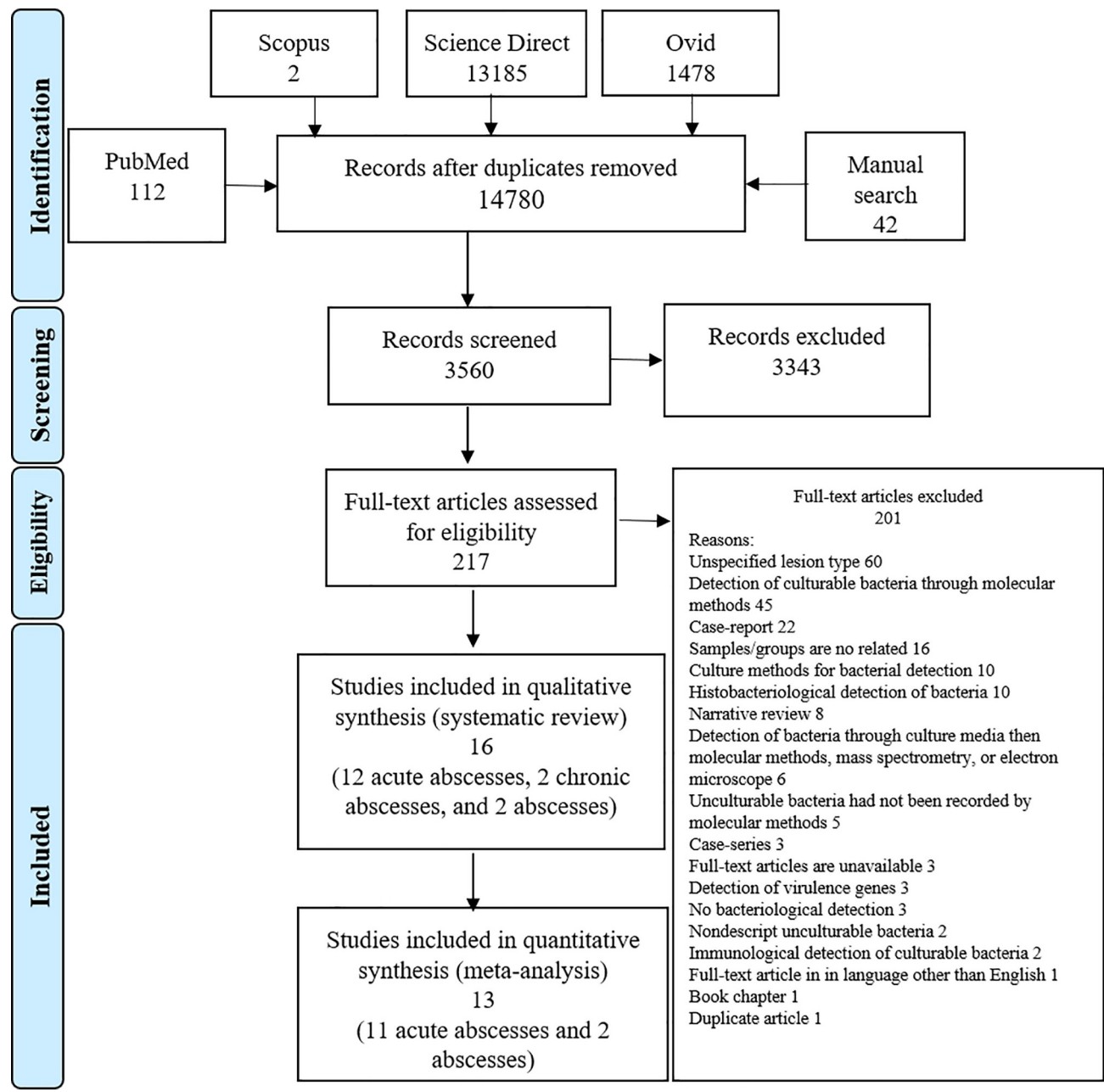

**Fig 1. PRISMA flowchart of the study selection process showing inclusion and exclusion of the identified studies.**

review [44, 46–60], from them, only 13 were included in the final quantitative meta-analysis [44, 46–52, 54–56, 59, 60] (Fig 1).

As none of the reviewed studies on periapical granuloma and radicular cyst reported the presence of unculturable bacteria, this systematic review and meta-analysis was confined only to the periapical abscess.

## Study characteristics

A total of 16 studies out of 14,780 studies (0.11%) were selected for the systematic review, and only 13 (0.09%) were used for the final analysis. All the included studies were cross sectional

study design and conducted between 2001 and 2020. Seven of the included studies were from Brazil [44, 50–54, 60], two from China [55, 56], two from USA [46, 47], and one each from South Korea [57], Republic of Estonia [59], UK [58], Brazil/USA [49], and Scotland [48]. The reviewed periapical lesions included 12 studies on acute periapical abscess [44, 46–56], two studies related to chronic periapical abscess [57, 58], while for the rest two studies [59, 60], the lesion type was recorded in our study as "periapical abscess" only without acute or chronic specification because one study mentioned the lesions as "abscess" [60], and the another represented an indeterminate analysis on acute/ chronic abscess [59] (Table 1).

Different molecular techniques were reported in regard to the detection of unculturable bacteria. Six (37.5%) out of 16 studies used the ABI SOLiD sequencing technology [44, 46, 49, 53, 55, 56], and two (12.5%) used terminal restriction fragment length polymorphism (T-RFLP) [48, 52]. Further, Illumine sequencer [59], pyrosequencing [57], denaturing high-performance liquid chromatography (dHPLC) [60], checker-board DNA-DNA hybridization [54], thermo sequenase sequencing [58], nested-polymerase chain reaction N-PCR [50], microbial array [47], and Semi-quantitative reverse-capture checkerboard assay [51] were used in only one study each (6.3%) (Table 1).

## Risk of bias assessment

The Critical Appraisal of Joanna Briggs Institute for Analytical Cross Sectional Studies [31] was used to evaluate the risk of bias. Cohen's kappa revealed that the observed agreement of inter-rater reliability for the risk of bias assessment was 94% and Cohen's kappa was 0.88 (95% CI: 0.64–1.00). The final scores of the included studies were determined after consensus process between the reviewers. All studies with a minimum score of 6/8 were included in the review. Of the 16 included studies, eight reached a score of 8/8 [51–53, 55–57, 59, 60], while the other eight got (6/8) [44, 46–50, 54, 58] of the bias assessment (S3 Table).

## Unculturable bacteria in periapical abscess

**Acute periapical abscess.** Of the 16 reviewed studies, 12 reported the presence of unculturable bacteria in acute periapical abscess [44, 46–56] (Table 2 and S4 Table). Briefly, in the acute periapical abscesses, 58 unculturable bacteria were identified from a total of 397 identified organisms (14.6%). Five studies reported both the abundance and frequency of isolated bacteria [44, 46, 52, 55, 56], and another five reported only the frequency of bacterial isolation [47, 49–51, 54], while only a single study provided the data on bacterial abundance [48].

There is a conflict in the literatures on the cultivability of the isolates from periapical abscess, as some reported the identical phylotypes are culturable, and others as unculturable. For instance, Siqueira and Rocas [53] reported Synergistes phylotype from periapical abscess as unculturable, but HOMD database indicates that all Synergistes sp., except for E3-33 E1, are culturable. Further, this study looked for only one phylotype of Synergistes and they did not mention neither the frequency nor the abundance of this unculturable bacterium and ultimately was excluded from the meta-analysis [53] (Table 1).

**Acute/chronic periapical abscess.** In general, only two studies [59, 60] revealed the prevalence of culturable and unculturable bacteria in the periapical abscesses without mentioning whether they are acute or chronic. For this reason, these studies were included in this systematic review and meta-analysis as "periapical abscess".

The total number of periapical abscess samples, unassigned as either acute or chronic were nine, and four unculturable isolates were reported out of a total 87 identified bacteria (4.6%) form these samples. Thus, Vengerfeldt et al., recorded the frequency of uncultured TG5 group as 25% [59]. Jacinto et al., reported the frequency of uncultured *Staphylococcus* sp. clone pGA

**Table 1. General characteristics of the included studies.**

| Author /year | Study design | Country | Sample size | Gender | Age | Dentition type | Number and lesion type | Sample collection procedure and origin | Molecular biology technique |
|---|---|---|---|---|---|---|---|---|---|
| Flynn et al. (2012) [46] | Cross sectional | USA | 9 | NM | NM | NM | 9 Acute periapical abscesses | Aspiration/Purulent | ABI |
| George et al. (2016) [47] | Cross sectional | USA | 18 | NM | 16–60 years | Permanent | 18 Acute periapical abscesses | Aspiration/Purulent | Microbial microarrays |
| Jacinto et al. (2007) [60] | Cross sectional | Brazil | 5 | NM | NM* | NM | 5 Periapical abscesses | Paper point/Root canal | Denaturing high-performance liquid chromatography |
| Riggio et al. (2007) [48] | Cross sectional | Scotland | 4 | 1 Woman, 3 men | 18-33years | Permanent | 4 Acute periapical abscesses | Surgical drainage/ Aspiration/Purulent | Terminal restriction fragment length polymorphism |
| Rôças and Siqueira (2009) [50] | Cross sectional | Brazil | 52 | NM | 18–74 years | Permanent | 21 Acute periapical abscesses | Aspiration/Purulent | Nested-PCR |
| Rôças and Siqueira (2018) [51] | Cross sectional | Brazil | 133 | 48 Women, 85 men | 16–75 years | Permanent | 55 Acute periapical abscesses | Aspiration/Purulent | Semi-quantitative reverse-capture checkerboard assay |
| Rôças et al. (2006) [49] | Cross sectional | Brazil, USA | 77 | NM | NM | Permanent | 77 Acute periapical abscesses | Aspiration/Purulent | ABI |
| Rolph et al. (2001) [58] | Cross sectional | UK | 41 | 17 Women, 8 men | NM | NM | 1 Chronic periapical abscess | Paper points/Root canal | Thermo Sequenase sequencing |
| Sakamoto et al. (2006) [52] | Cross sectional | Brazil | 16 | NM | 18–44 years | Permanent | 7 Acute periapical abscesses | Aspiration/Purulent | Terminal restriction fragment length polymorphism |
| Sakamoto et al. (2009) [44] | Cross sectional | Brazil | 90 | NM | NM | Permanent | 6 Acute periapical abscesses | Aspiration/Purulent | ABI |
| Siqueira and Rôças (2007) [53] | Cross sectional | Brazil | 50 | NM | >18 years | Permanent | 29 Acute periapical abscesses | Aspiration/Purulent | ABI |
| Siqueira and Rôças (2009) [54] | Cross sectional | Brazil | 42 | NM | NM* | Permanent | 42 Acute periapical abscesses | Aspiration/Purulent | Checkerboard DNA-DNA hybridization |
| Vengerfeldt et al. (2014) [59] | Cross sectional | Republic of Estonia | 12 | 7 Men, 5 women | 27–66 years | Permanent | 3 Acute and 1 chronic periapical abscess (periapical abscesses) | Paper points/Root canal | Illumina Sequencing |
| Yang et al. (2010) [55] | Cross sectional | China | 11 | 5 Boys, 6 girls | 5.4–7.6 years | Deciduous | 11 Acute periapical abscesses | Puncturing the mucosa and paper points inserted into the mucosa | ABI |
| Yun et al. (2017) [57] | Cross sectional | South Korea | 10 | NM | 2–7 years | Deciduous | 2 Chronic periapical abscesses | Paper points/Root canal | Pyrosequencing |
| Zhang et al. (2020) [56] | Cross sectional | China | 9 | 6 Boys, 3 girls | 3–11 years | Deciduous | 9 Acute periapical abscesses | Paper point/Root canal | ABI |

NM: not mentioned, NM*: not mentioned but checked by the authors.

2 as 40%; while 20% for the uncultured rape rhizosphere bacterium wr0200 and bacterium clone aab38e07 was noted among the periapical abscess samples [60]. Jacinto et al., also recorded the abundance of the uncultured bacterial clones among all other bacterial clones. The uncultured rape rhizosphere bacterium wr0200 was recorded as the most abundant 2.71%, while the *Staphylococcus* sp. clone pGA 2 and uncultured bacterium clones aab38e07 were 2.08% and 1.46%, respectively [60] (S4 Table).

**Table 2. Extracted results from each study of periapical abscess.**

| Study | Number of unculturable bacteria | Total number of bacteria | Diversity % | Clones of unculturable bacterial | Total number of bacterial clones | Abundance % | Frequency of unculturable bacteria | Total frequency of bacteria | Frequency % |
|---|---|---|---|---|---|---|---|---|---|
| Flynn et al. (2012) [46] | 5 | 25 | 20.00 | 6 | 391 | 1.53 | 6 | 67 | 8.96 |
| George et al. (2016) [47] | 4 | 41 | 9.76 | _ | _ | _ | 54 | 550 | 9.82 |
| Riggio et al. (2007) [48] | 3 | 29 | 10.34 | 12 | 203 | 5.91 | _ | _ | _ |
| Rocas et al. (2006) [49] | 1 | 10 | 10.00 | _ | _ | _ | 2 | 267 | 0.75 |
| Rocas and Siqueira (2009) [50] | 1 | 3 | 33.33 | _ | _ | _ | 3 | 9 | 33.33 |
| Rocas and Siqueira (2018) [51] | 2 | 39 | 5.13 | _ | _ | _ | 11 | 455 | 2.42 |
| Sakamoto et al. (2006) [52] | 9 | 29 | 31.03 | 24 | 93 | 25.81 | 18 | 85 | 21.18 |
| Sakamoto et al. (2009) [44] | 15 | 24 | 62.50 | 18 | 287 | 6.27 | 16 | 37 | 43.24 |
| Siqueira and Rocas (2009) [54] | 11 | 55 | 20.00 | _ | _ | _ | 39 | 354 | 11.02 |
| Yang et al. (2010) [55] | 1 | 17 | 5.88 | 8 | 424 | 1.89 | 1 | 62 | 1.61 |
| Zhang et al. (2020) [56] | 6 | 125 | 4.80 | _ | _ | _ | _ | _ | _ |
| Jacinto et al. (2007) [60] | 3 | 33 | 9.09 | 30 | 480 | 6.25 | 4 | 48 | 8.33 |
| Vengerfeldt et al. (2014) [59] | 1 | 54 | 1.85 | _ | _ | _ | 1 | 136 | 0.74 |
| **Total** | **62** | **484** | | **98** | **1878** | | **155** | **2070** | |

**Chronic periapical abscess.** The uncolturable bacteria from chronic periapical abscess were reported in two studies [57, 58]. Collectively, three samples of chronic periapical abscesses were reported and two unculturable bacteria were identified from a total of 23 culturable bacteria (8.7%). The chronic abscesses were samples from primary [57] and permanent teeth [58]. Yun et al., reported that the abundance of uncultured bacterium AF287795_g (*Selenomonas* sp. oral clone CS002/ *Mitsuokella* sp. Human Oral Taxon (HOT) 131) is 16% [57]. On the other hand, Rolph et al., reported the unidentified *Eubacterium* clone 3.3 [U43698] in one case of chronic periapical abscess, although its isolation frequency and the abundance were not reported [58]. These two studies [57, 58] were excluded from the meta-analysis due to low sample size as one of them included two cases [57], and the other study included only one case of chronic periapical abscesses [58].

## Outcome measures: Primary outcome measures: Diversity, abundance, and frequency of unculturable bacteria in periapical abscess

In this meta-analysis, the diversity, abundance, and frequency of unculturable bacteria were studied. The total number of unculturable bacteria reported in all abscess samples was 62 out of a total 484 identified bacteria. The diversity of unculturable bacteria in periapical abscesses

[59, 60] and in acute periapical abscess [44, 46–52, 54–56] were determined. The diversity of unculturable bacteria in all abscess samples was 13%, 95% CI: 7, 23% (Fig 2A).

The abundance of unculturable bacteria in periapical abscess [60] and in acute periapical abscess [44, 46–48, 52, 55] were calculated. The estimate of the pooled abundance of unculturable bacteria in all abscess samples was 5%; 95% CI: 2, 12% (Fig 2B).

The frequency of unculturable bacteria in periapical abscess [59, 60] and in acute periapical abscess [44, 46, 47, 49–52, 54, 55] were determined. The pooled frequency of unculturable bacteria in all abscess samples was 8%; 95% CI: 5, 14% (Fig 2C).

**Heterogeneity of studies.** The heterogeneity between studies were evaluated by meta-regression analysis. For this purpose, we estimated which specific moderator had a significant impact on the outcomes using three arbitrary categories, depending on the availability of data. Namely, i) the type of lesion, ii) the country of study origin, and iii) the sequencing technique employed. Lesion moderation was deemed dichotomous, as acute abscess, and abscess. Similarly, in the country of origin moderation, two divisions were evaluated, Brazil and other countries, as most data originated from Brazil, while for "sequence" moderation, the quality of DNA sequencing and hybridization were used as two separate categories. The term "sequence" represented a nonspecific 16S rRNA targeted gene sequencing of bacterial species using techniques such as ABI, pyrosequencing, Illumina sequencing, and thermo sequencing, while the "hybridization" refers to a specific 16S rRNA targeted gene identification using techniques such as T-RFLP, N-PCR, microbial microarrays, dHPLC, checkerboard DNA-DNA hybridization, and semi-quantitative reverse-capture checkerboard assay.

For diversity estimate, the heterogeneity ($I^2$) was high (78%, 95% CI: 63, 87%), with $tau^2$ = 1.09/0.32; 3.5, and $P < 0.01$. As for lesion moderation, there were two studies denoting the lesions as "abscess" [59, 60], and 11 studies defined as "acute abscess" [44, 46–52, 54–56]. The heterogeneity ($R^2$) computed for the latter was 3.86%, the test of moderator coefficient was [QM (df = 1) = 1.74, P = 0.19], and the significant slope coefficient was 1.30; Z (15) = 1.32, and P = 0.19. All the meta-regression analyses revealed that the lesion moderation did not significantly impact the summary proportions (S1A Fig). With the country moderation, the heterogeneity was 33.83%, the test of moderator coefficient was [QM (df = 1) = 3.86, P = 0.05], and the significant slope coefficient was 1.17; Z (15) = 1.96, and P = 0.05, representing a significant effect on summary proportion (S1B Fig). Regarding the sequence technique moderator, the amount of heterogeneity ($R^2$) was 0.00%, the test of moderator coefficient was [QM (df = 1) = 0.04, P = 0.85], and the significant slope coefficient was -0.14; Z (15) = -0.19, and P = 0.85, depicting no significant effect of the sequence techniques on the summary proportion (S1C Fig).

For abundance estimate, the amount of heterogeneity ($I^2$) was high (93%, 95% CI: 87.6, 96.1%), $tau^2$ = (0.96/0.36; 6.99), and $P < 0.01$. Meta-regression analysis was conducted to estimate which moderator had a considerable impact on the abundance summary proportion. With the lesion moderator, the abscess lesion was recorded in one study [60], and acute abscess in five studies [44, 46, 48, 52, 55]. The amount of heterogeneity ($R^2$) was 0.00%, the test of moderator coefficient was [QM (df = 1) = 0.02, P = 0.9], and the significant slope coefficient was -0.19; Z (15) = -0.14, and P = 0.9 (S2A Fig). Regarding the country moderator, the amount of heterogeneity ($R^2$) was 25.14%, the test of moderator coefficient was [QM (df = 1) = 3.86, P = 0.05], and the significant slope coefficient was 1.44; Z (15) = 1.96, and P = 0.05 (S2B Fig). For the sequence methods moderator, the amount of heterogeneity ($R^2$) was 15.46%, the test of moderator coefficient was [QM (df = 1) = 3.26, P = 0.07], and the significant slope coefficient was -1.4; Z (15) = -1.8, and P = 0.07 (S2C Fig).

For frequency, the amount of heterogeneity ($I^2$) was high (89.9%, 95% CI: 83.9, 93.6%), $tau^2$ = (0.87 /0.58; 5.49), and ($P < 0.01$). In meta-regression analysis, the lesion moderator for

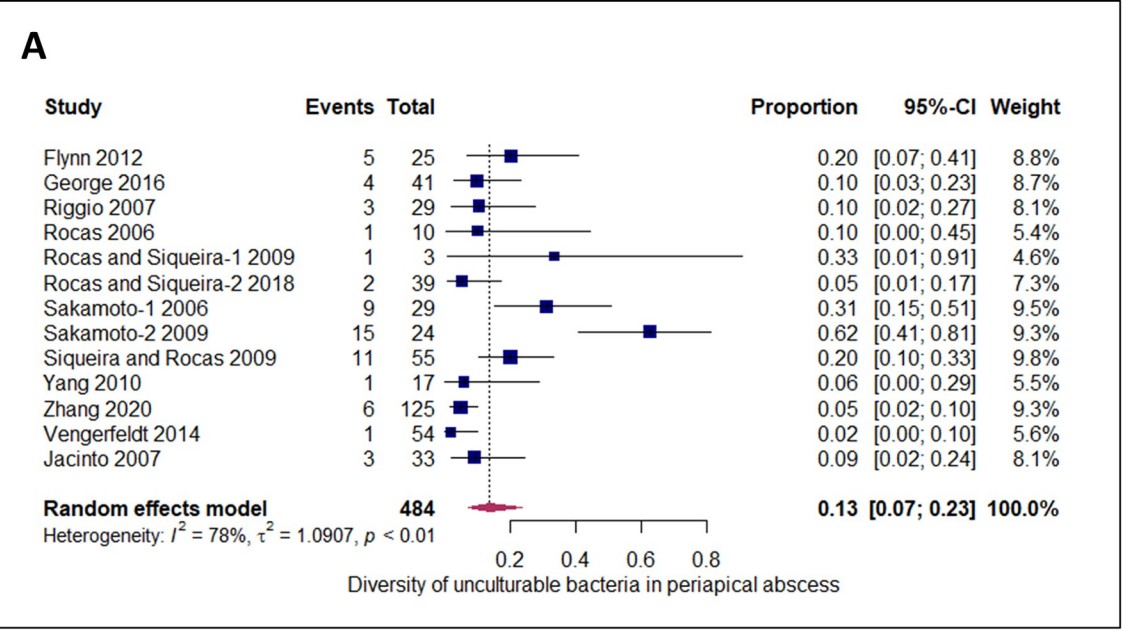

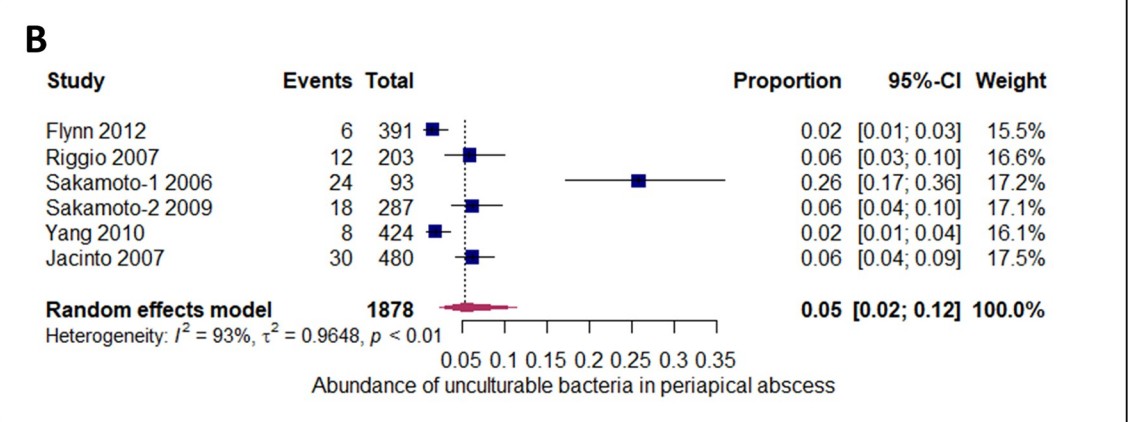

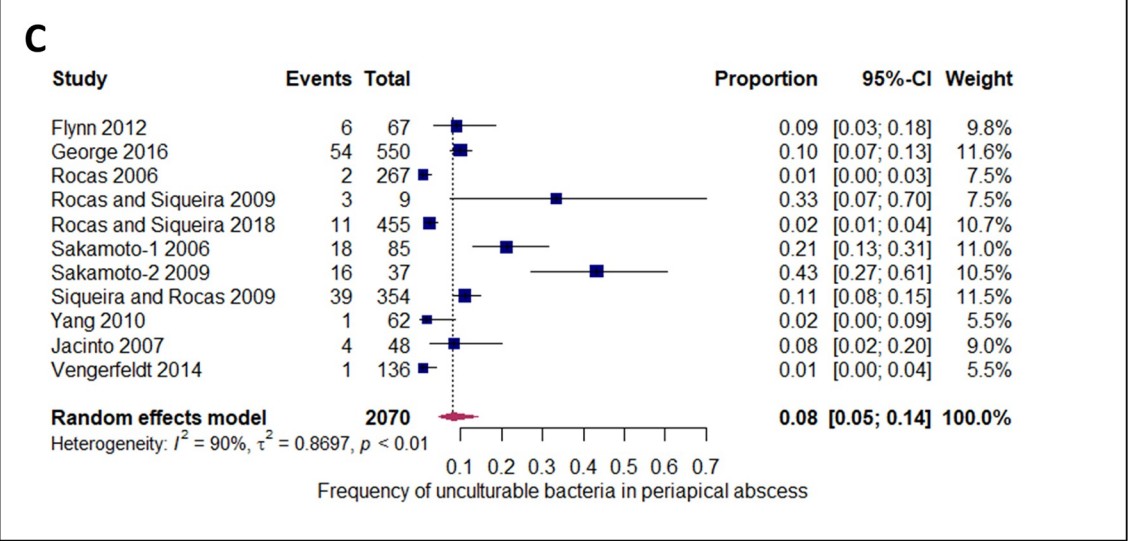

**Fig 2. Forest plot depicting the pooled estimate of unculturable bacteria in periapical abscess.** (A) Diversity of unculturable bacteria in periapical abscess was estimated to be 13% (95%, CI: 7, 23%) in 13 studies. (B) Abundance of unculturable bacteria was estimated to be 5% (95%, CI: 2, 12%) in six studies. (C) Frequency of unculturable bacteria in periapical abscess was estimated to be 8% (95%, CI: 5, 14%) in 11 studies.

abscess studies [59, 60] and acute abscess studies [44, 46, 47, 49–52, 54, 55] revealed that the amount of heterogeneity ($R^2$) was 0.00%, the test of moderator coefficient was [QM (df = 1) = 1.41, P = 0.24], and the significant slope coefficient was 1.09; Z(15) = 1.18, and P = 0.24. These results indicated that the lesion moderator did not significantly affect the summary proportion (S3A Fig). For the country moderator, the amount of heterogeneity ($R^2$) was 0.00%, the test of moderator coefficient was [QM (df = 1) = 1.44, P = 0.23], and the significant slope coefficient was 0.96; Z (15) = 1.21, and P = 0.23 (S3B Fig). In the case of sequence methods moderator, the amount of heterogeneity ($R^2$) was 0.00%, the test of moderator coefficient was [QM (df = 1) = 1.3, P = 0.25], and the significant slope coefficient was -0.78; Z (15) = -1.14, and P = 0.25 (S3C Fig).

**Outlying studies.**    Several analyses were performed for detection the outlying studies. For the diversity of unculturable bacteria in periapical abscess, studentized residuals tests showed that study number 8 [44] has an outlying z-value 3.05. Leaving out this study [44] revealed some changes on the summary proportion from 13%; 95% CI: 7, 23% to 11%; 95% CI: 7, 18% (S4A Fig). Externally studentized residuals (rstudent) test, revealed that all studies are normally distributed except for the 8th study which is located outside the range limit [44]. DFFITS values showed the standard deviations of the 8th study [44] was the highest (0.82). Also, Cook's distances (cook.d) was 0.34 for the 8th study [44] and co-variance ratios (cov.r) was 0.62. The amount of heterogeneity ($tau^2$.del) displayed $tau^2$ after the omitting of the 8th study [44] was 0.46. The test statistics for heterogeneity (QE.del) was 26.61 if the 8th study [44] is omitted, hat values was 0.09, and lastly the weights was 9.33. All these tests confirmed that study [44] is an outlying study but not influential, so it was not removed (S4B Fig).

For abundance estimate, the outlying studies, studentized residuals tests and leaving out each study revealed that the study number 3 [52] is an outlier with a z-value of 3.23, and summary proportion has changed from 5%; 95% CI: 2, 12% to 4%; 95% CI: 2, 7% after removing the 3rd study [52] (S5A Fig). The rstudent test was 3.23, DFFITS was 1.33, cook.d test was 0.63, cov. r test was 0.45, tau2.del test was 0.3, QE.del test was 19.52, the hat and weight tests were 0.17 and 17.18 after removing the 3rd study [52], respectively. Although these tests reveal that the 3rd study [52] is an outlier and influential study (S5B Fig), we did not exclude it because its effect was minor on abundance summary proportion to be only from 5% to 4% when leaving out this study (Fig 1B and S5A Fig).

For frequency estimate, studentized residuals tests showed that the study number 7 [44] has an outlying z-value 2.62. Leaving out this study revealed some changes on the summary proportion from 8%; 95% CI: 5, 14% to 7%; 95% CI: 4, 11% (S6A Fig). The rstudent test was 2.62, DFFITS was 0.77, cook.d test was 0.44, cov. r test was 0.85, tau2.del test was 0.61, QE.del test was 64.98, the hat and weight tests were 0.11 and 10.52 after removing the 7th study [44]. The results confirmed that study [44] is an outlying study but did not have an influential effect (S6B Fig).

**Publication bias of studies.**    The publication bias was determined visually through the funnel plot, and statistically, by the unweighted Egger's regression test. S7A Fig showed the funnel plot of diversity estimate of unculturable bacteria and Egger's regression test was z = -1.2905, and P = 0.1969, which revealed non-significant publication bias. For the abundance, Egger's regression test was z = -1.92, and P = 0.06, which showed no significant publication bias (S7B Fig). On the other hand, funnel plot and Egger's regression test showed a significant

publication bias for the frequency estimate of unculturable bacteria (z = -2.17, and P = 0.03) (S7C Fig).

## Outcome measures: Secondary outcome measures: Abundance and frequency of individual unculturable bacteria in periapical abscess

In periapical abscess, the abundance and frequency of unculturable bacteria were determined (S4 Table). The secondary outcomes were measured for 35 unculturable bacteria using three variables, the abundance and frequency as parametric variables and the sequence techniques as non-parametric variable (Table 3). PCA was performed, where the abundance and frequency were represented as active variables, while the sequence techniques as a supplementary-categorical variable. PCA for individuals (unculturable bacteria) and for variables have been calculated and visualized using biplot (Fig 3). The variance percentages were 60.04 and 39.96 for PC1 and PC2, respectively (Fig 3). Regarding the PCA for the individual unculturable bacteria, the acute angle between the abundance and frequency variables and the Pearson's correlation between the abundance and frequency revealed a positive but non-significant correlation (0.2, P = 0.3) (Fig 3 and S8 Fig).

For better interpretation, bacteria were represented as numbers (Table 3). Biplot showed that bacterium 11 has the highest abundance and frequency followed by 9, 7, and 33. The other bacteria including 13, 10, 34, 15, 12, and 31 represented the second highest abundances in a descending order, while bacteria 10, 15, 13, 12, 31, and 34 considered as the second highest frequency. Conversely, 1, 2, and 5 showed the lowest abundance and with similar values. Bacteria 4, and 32 showed the second lowest abundance, while bacteria 16, 18, 21, 22, 23, 24, 25, 26, 27, 28, and 29 were the third lowest abundance. On the other hand, the frequency of bacteria 1, 2, and 5 were the lowest, while bacteria 4, and 32 were the second lowest frequency and bacterium 30 was the third. Bacteria 16, 18, 21, 22, 23, 24, 25, 26, 27, 28, 29, 6, 8, and 14 showed the fourth lowest frequency. The other bacteria including 3, 17, 19, 20, and 35 were nearly the same average for both abundance and frequency (Fig 3).

The supplementary variable, sequence techniques, was employed for grouping the bacteria into two major groups, "sequence" or "hybridization" using confidence ellipse (Fig 4). The confidence ellipse of hybridization group was 1, -0.01, while the confidence ellipse of sequence group was -0.5, 0.01, indicating that the hybridization technique is more prominent.

## Discussion

To the best of our knowledge, this is the first systematic review and meta-analysis on unculturable bacteria in, periapical abscess, periapical granulomas or cysts. In our study, the outcomes of the included studies on the identification of unculturable bacteria generally provided information about the diversity, abundance, and frequency of the bacterial phylotypes. Molecular identification have shown that the bacterial diversity in most environments is strongly underestimated in culture-based techniques [17, 61] and 34% of the identified bacterial taxa are uncultivated [62], indicating that these unrecognized bacterial species present in the periapical lesions may participate in persistent apical periodontitis [62], periapical abscess [63] or the pathogenesis of other oral diseases [17, 18, 64]. We noted that the diversity score of unculturable bacteria in the acute periapical abscess was 13%. This figure is different from previous studies where the diversity percentage of unculturable bacteria in periapical abscess was 24% [46] while in another study it was 55% [13]. On the other hand, a low diversity score (2%) was noted in one study where the lesion was simply defined as a 'dental abscess' without referring to the acute/chronic nature of the lesion [59]. This discrepancy in the diversity score could depend on the country where the experiments were conducted, as for instance, we noted a

**Table 3. Secondary outcome measures of individual unculturable bacteria with active abundance and frequency, and supplementary sequence technique variables.**

| Symbol number of bacteria | Studies of acute periapical abscess | Unculturable bacteria | Abundance % | Frequency % | Sequence techniques |
|---|---|---|---|---|---|
| 1 | Flynn et al. (2012) [46] (Abundance/ Frequency) | *Leptotrichia* sp. AM420283 | 0.26 | 11.11 | Sequence |
| 2 | | *Leptotrichia* [G-1] sp. Oral Taxon 220 | 0.26 | 11.11 | Sequence |
| 3 | | *Peptostreptococcaceae* [G-4] sp. Oral Taxon 103 | 0.51 | 22.22 | Sequence |
| 4 | | *Synergistes*[G-3] sp. Oral Taxon 360 | 0.26 | 11.11 | Sequence |
| 5 | | *Prevotella* sp. Oral Taxon 315 | 0.26 | 11.11 | Sequence |
| 6 | Sakamoto et al. (2006) [52] (Abundance/Frequency) | Bacteroidales oral clone MCE7_164/MCE3_262/MB4_G15 | 1.08 | 14 | Hybridization |
| 7 | | *Prevotella* sp. E9_42/*Preqotella* sp. oral clone PUS9.180 | 2.15 | 43 | Hybridization |
| 8 | | Uncultured Eubacterium E1-K13 | 1.08 | 14 | Hybridization |
| 9 | | Lachnospiraceae oral clone 55A-34 | 1.08 | 57 | Hybridization |
| 10 | | Lachnospiraceae oral clone MCE7-60 | 1.08 | 43 | Hybridization |
| 11 | | *Peptostreptococcus* sp. oral clone CK035 | 11.8 | 29 | Hybridization |
| 12 | | *Selenomonas* sp. oral clone 55A-7 | 1.08 | 29 | Hybridization |
| 13 | | Bacterium MDA2477/Bacterium MDA2477-like oral clone 51A-9 | 5.38 | 14 | Hybridization |
| 14 | | *Pseudomonas* sp. LCY11 | 1.08 | 14 | Hybridization |
| 15 | Sakamoto et al. (2009) [44] (Abundance/Frequency) | *Treponema* sp. oral taxon IV:18:C9 | 0.70 | 33.3 | Sequence |
| 16 | | Treponema clone 142–10 | 0.35 | 16.7 | Sequence |
| 17 | | Treponema clone 142–21 | 0.70 | 16.7 | Sequence |
| 18 | | Treponema clone 142–82 | 0.35 | 16.7 | Sequence |
| 19 | | Treponema clone 18f-1 | 0.70 | 16.7 | Sequence |
| 20 | | Treponema clone 18f-6 | 0.35 | 16.7 | Sequence |
| 21 | | Treponema clone 18f-7 | 0.35 | 16.7 | Sequence |
| 22 | | Treponema clone 18f-22 | 0.35 | 16.7 | Sequence |
| 23 | | Treponema clone 18f-33 | 0.35 | 16.7 | Sequence |
| 24 | | Treponema clone 18f-35 | 0.35 | 16.7 | Sequence |
| 25 | | Treponema clone 18f-48 | 0.35 | 16.7 | Sequence |
| 26 | | Treponema clone 94A-72 | 0.35 | 16.7 | Sequence |
| 27 | | Treponema clone 94A-89 | 0.35 | 16.7 | Sequence |
| 28 | | Treponema clone 94A-92 | 0.35 | 16.7 | Sequence |
| 29 | | Treponema clone 94A-94 | 0.35 | 16.7 | Sequence |
| 30 | Yang et al. (2010) [55] (Abundance/Frequency) | Bacteroidales genomosp. P4 oral clone MB2_G17 | 1.9 | 9.1 | Sequence |
| 31 | Zhang et al. (2020) [56] (Abundance/Frequency) | *Acinetobacter* sp. Oral taxon 408 | 0.75 | 33 | Sequence |
| 32 | | *Peptostreptococcaceae* [XI][G-7] sp. oral taxon 081 | 0.56 | 11 | Sequence |
| 33 | Jacinto et al. (2007) [60] (Abundance/Frequency) | Uncultured *Staphylococcus* sp. clone pGA 2 | 2.08 | 40 | Hybridization |
| 34 | | Uncultured rape rhizosphere bacterium wr0200 | 2.71 | 20 | Hybridization |
| 35 | | Uncultured bacterium clone aab38e07 | 1.46 | 20 | Hybridization |

significant higher diversity score in Brazil when compared to other countries (P = 0.05) (S1B Fig). In addition, it has been reported previously that geographical locations have an impact on the bacterial community [65, 66] and hence bacterial diversity. Furthermore, the molecular methods used in the studies may influence the sensitivity of unculturable phylotypes. For

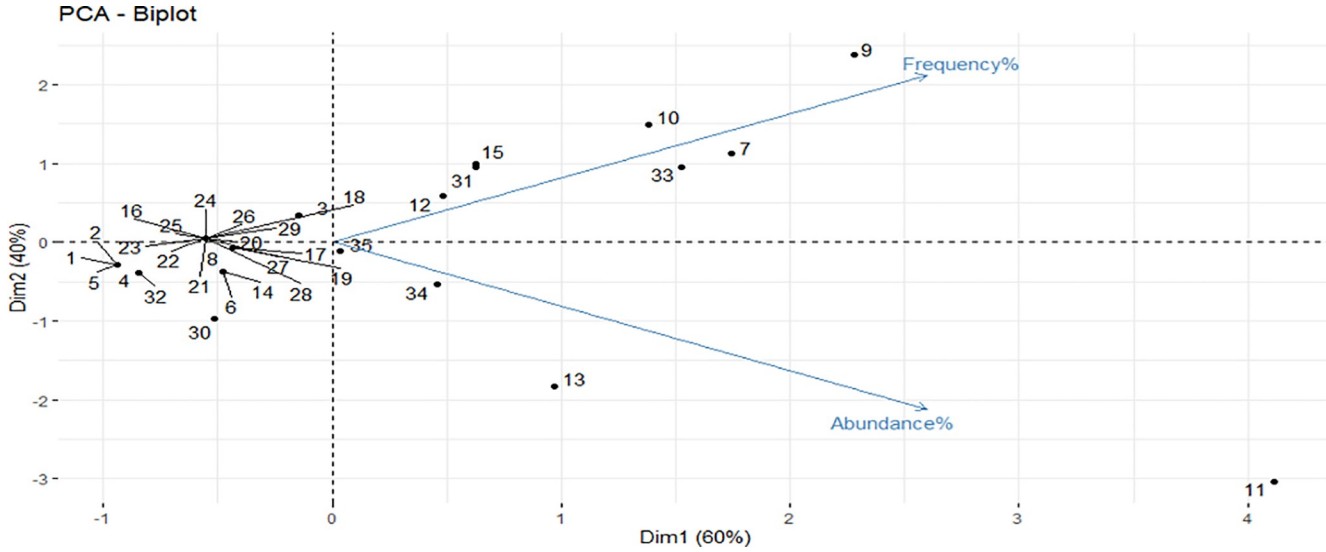

**Fig 3. Biplot of abundance and frequency dimensions of individual unculturable bacteria in periapical abscesses.**

example, a study reported the diversity score of 55% when using T-RFLP (hybridization technique) [13], while another one reported this figure as 2% on using the proprietary Illumina sequencing method [59]. The others have noted that the increasingly advanced techniques used to detect the unculturable bacterial phylotypes may affect the diversity score due to newly included phylotypes of unculturable bacteria [67–70].

Bacterial abundance and their interactions are significantly correlated to the severity of acute periapical abscesses [51, 65]. However, in term of abundance, different teeth with acute

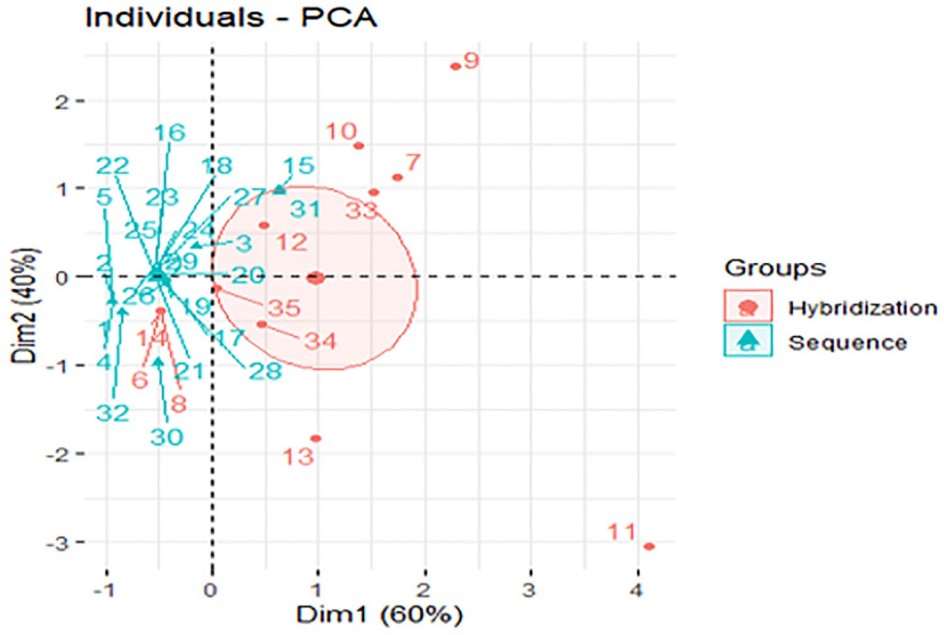

**Fig 4. PCA of individual unculturable bacteria in periapical abscesses categorized by sequence techniques supplementary variable.**

apical periodontitis represent heterogeneous bacterial abundance [71, 72]. In the current meta-analysis, the abundance of unculturable bacterial phylotypes was computed to be 5%, concurring with a study noted a similar abundance of 6% [48]. On the other hand, a higher abundance of 26% was found [13], and a lower percentages of 1.5% [46] and 2% [55] were also reported. We also noted a significant difference the abundance of phylotypes depending on the country of study origin. For instance, studies performed in Brazil [44, 52, 60] showed a higher phylotype abundance when compared to those conducted in other countries [46, 48, 55] (S2B Fig). One possible explanation for this finding is the variation in analytical techniques used in different jurisdictions. Thus, more unculturable clones were detected in Brazilian studies using T-RFLP analysis [13, 48] in contrast to lower detection rates noted by others using ABI sequencing method [46, 55].

The frequency of bacterial species and its correlation to periapical lesions was studied for the first time in 2018 [51]. This study revealed a significant correlation between the frequency of specific bacterial taxa and the asymptomatic form of apical periodontitis [51]. The frequency of unculturable bacteria in periapical abscess was 8% in our study. This figure is intermediate with regards to previous data where both a low frequency of 1–2% [49, 51, 55, 59], and higher frequencies of 43% [44] and 33% [50] have been noted. Significant publication bias was detected during the identification of the frequency of unculturable bacteria in periapical abscess. Frequency is positively correlated to diversity [73] and the type of sequencing methods may have an impact on the frequency. According to our results, we found that the high frequency of unculturable bacteria in some studies were related to their high diversity in the same studies and vice versa (Table 2). Furthermore, the significant publication bias in frequency can be due to the different approach for sequencing. In our study, the lowest frequencies were detected in three studies [49, 55, 59] that used sequencing methods, while the highest frequencies were detected in another study [50] that used the hybridization technique. Although another highest frequency was detected by Sakamoto et al. [44], this study focused on a specific genus (*Treponema*), for that reason the frequency was high despite using a sequencing technique.

Regarding the identity of unculturable bacteria, their abundance could be positively correlated with the identification frequency. An exponential increase in the abundance and frequency of unculturable bacteria, *Leptotrichia* sp. AM420283 and *Peptostreptococcaceae* [G-4] sp. Oral Taxon 103, was reported [46]. Another study [52] also reported a high frequency of isolation of *Lachnospiraceae* oral clone 55A-34, *Lachnospiraceae* oral clone MCE7_60, and *Selenomonas* sp. oral clone 55A-7 to be 57%, 43%, and 29%, respectively. While the abundance was 1.8% for all the foregoing bacteria.

In the present study, the highest variable for abundance and frequency of unculturable bacteria was reported for *Peptostreptococcus* sp. oral clone CK035. Accordingly, a study found that its abundance was 11.8% in acute periapical abscess [52] compared to 15% in chronic periapical lesions [74]. The first detection of this bacterium was figured out in chronic apical periodontitis with 25% for its frequency [75].

Confidence ellipse of hybridization and sequence of individual unculturable bacteria clearly separated the bacteria into two groups according to their values of abundance and frequency. As shown in Fig 4, the hybridization techniques used, including T-RFLP, N-PCR, microbial microarrays, dHPLC, checkerboard DNA-DNA hybridization, or semi-quantitative reverse-capture checkerboard assay appear to be more powerful and sensitive tools for recording the total number of clones than the sequencing techniques including ABI, pyrosequencing, illumina sequencing, or thermos-sequenase sequencing. Comparing the T-RFLP with illumina sequencing method [76, 77] or with Ion Torrent PGM [78] revealed that T-RFLP is a powerful and highly reliable method for microbial screening. However, further studies are needed to

confirm this result specifically if the aim of the study was targeted towards low or moderate abundant bacteria like unculturable phylotypes as shown in our study.

In this systematic review and meta-analysis, the unculturable bacteria in periapical lesions including periapical abscess, granuloma, and cyst were investigated. Only one article used a molecular method in detecting organisms in both the periapical granulomas and cysts, but only to the genus level of the isolates [79]. Some studies focused on a specific, culturable or unculturable phylotypes of a bacterial genus, while ignoring the in-depth screening of other bacterial genera or unculturable phylotypes [44]. The review indicates that there is a dearth of research on unculturable bacteria in periapical granuloma and radicular cysts. Therefore, more research is warranted on chronic periapical abscess, periapical granuloma, and radicular cyst with an emphasize on the diversity, abundance, and frequency of the unculturable bacteria. This will allow us to understand the role of these newly described organisms in the disease process and pave the way for more effective and efficacious therapeutic approaches not only for periapical lesions but also for other infections in general.

## Conclusion

This review indicates that unculturable bacteria are moderately exist in the periapical abscess and such prevalence may significantly contribute to their pathogenesis. However, their role in the pathogenesis is yet to be determined since their prevalence and abundance may not necessarily reflect their activity. Some organisms may be dormant or inactive while others may be highly active. Nevertheless, we noted a positive correlation between the abundance and frequency of individual unculturable bacterium *Peptostreptococcus* sp. oral clone CK035 in periapical abscesses. In terms of methodology, hybridization techniques appear to be more reliable in detecting the abundance and frequency of individual unculturable bacteria in periapical lesions. Collectively, careful analysis of the available data regarding the uncultivable bacteria indicates their importance in the progression of periapical abscess and hence therapeutic strategy, while more experimental screening is still required.

## Supporting information

**S1 Checklist. PRISMA 2009 checklist.**
(DOC)

**S1 Table. Literature search strategy and terms for the prevalence of unculturable bacteria in periapical lesions.**
(PDF)

**S2 Table. Checklist of critical appraisal of Joanna Briggs Institute for analytical cross sectional studies.**
(PDF)

**S3 Table. Bias assessment of included studies using Joanna Briggs Institute for analytical cross sectional studies.**
(PDF)

**S4 Table. Extracted results from each study of individual bacteria in periapical abscess.**
(PDF)

**S1 Fig. Scatter plot depicting the moderators' effect on summary proportion of diversity of unculturable bacteria in periapical abscess.** (A) Lesion moderator. (B) Country moderator. (C) Sequence technique moderator.
(TIF)

**S2 Fig. Scatter plot depicting the moderators' effect on summary proportion of abundance of unculturable bacteria in periapical abscess.** (A) Lesion moderator. (B) Country moderator. (C) Sequence technique moderator.
(TIF)

**S3 Fig. Scatter plot depicting the moderators' effect on summary proportion of frequency of unculturable bacteria in periapical abscess.** (A) Lesion moderator. (B) Country moderator. (C) Sequence technique moderator.
(TIF)

**S4 Fig. Results for outlying studies of diversity of unculturable bacteria in periapical abscess.** (A) Forest plot depicting summary proportion after leaving out each study. (B) Influential analysis plot of diversity of unculturable bacteria in thirteen studies of periapical abscess.
(TIF)

**S5 Fig. Results for outlying studies of abundance of unculturable bacteria in periapical abscess.** (A) Forest plot depicting summary proportion after leaving out each study. (B) Influential analysis plot of abundance of unculturable bacteria in six studies of periapical abscess.
(TIF)

**S6 Fig. Results for outlying studies of frequency of unculturable bacteria in periapical abscess.** (A) Forest plot depicting summary proportion after leaving out each study. (B) Influential analysis plot of frequency of unculturable bacteria in eleven studies of periapical abscess.
(TIF)

**S7 Fig. Funnel plot of publication bias of unculturable bacteria in periapical abscess.** (A) Diversity. (B) Abundance. (C) Frequency.
(TIF)

**S8 Fig. Scatter plot of abundance and frequency correlation for individual unculturable bacteria.**
(TIF)

## Author Contributions

**Conceptualization:** Alaa Muayad Altaie, Lakshman P. Samaranayake.

**Data curation:** Alaa Muayad Altaie, Mohammed Amjed Alsaegh.

**Formal analysis:** Alaa Muayad Altaie.

**Investigation:** Alaa Muayad Altaie, Mohammed Amjed Alsaegh, Lakshman P. Samaranayake.

**Methodology:** Alaa Muayad Altaie, Basema Saddik, Lakshman P. Samaranayake.

**Project administration:** Basema Saddik, Sameh S. M. Soliman, Rifat Hamoudi, Lakshman P. Samaranayake.

**Supervision:** Sameh S. M. Soliman, Rifat Hamoudi, Lakshman P. Samaranayake.

**Validation:** Basema Saddik, Sameh S. M. Soliman, Rifat Hamoudi, Lakshman P. Samaranayake.

**Writing – original draft:** Alaa Muayad Altaie.

**Writing – review & editing:** Basema Saddik, Mohammed Amjed Alsaegh, Sameh S. M. Soliman, Rifat Hamoudi, Lakshman P. Samaranayake.

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
