## [Decision Letter · Decision Letter 0]

21 May 2021

PONE-D-20-39104

Prevalence of unculturable bacteria in the dentoalveolar abscess and related periapical lesions: A systematic review and meta-analysis

PLOS ONE

Dear Dr. Hamoudi,

First, please accept our apologies for having waited so long for our decision on your submitted draft. Unfortunately, the original Editor was, and I have stepped in as the Academic Editor some few hours ago.Thank you for submitting your manuscript to PLOS ONE. After careful consideration, we feel that it has merit but does not fully meet PLOS ONE’s publication criteria as it currently stands. Therefore, we invite you to submit a revised version of the manuscript that addresses the points raised during the review process.Having intensively reviewed your draft, several minor and major shortcomings have been identified by our external experts. Therefore, I have double checked your submission, to come to a more balanced decision. No doubt, the concerns indicated by our reviewers cannot be ignored. Thus, after deliberate consideration, the Editorial Board feels that your manuscript's quality should be increased prior to re-entering into a further review process, to forward an improved draft of your work to further external reviewers. All in all, the indicated shortcomings are considered reasonable with regard to both PLOS ONE's quality standards and our readership's expectations, and another round of revisions would seem mandatory. Please note that your re-submitted draft will again be forwarded to re-review.Therefore, we invite you to submit a thoroughly revised version of the manuscript that addresses EACH AND EVERY point raised during this review process. Please note that another non-convincing revision (not considered acceptable with regard to language, content, reviewers' constructive criticism, generalizable conclusions, and/or Authors' Guidelines) will lead to outright reject.Please submit your revised manuscript by Jul 01 2021 11:59PM. If you will need more time than this to complete your revisions, please reply to this message or contact the journal office at plosone@plos.org. Please include the following items when submitting your revised manuscript:- A rebuttal letter that responds to each point raised by the academic editor and reviewer(s). You should upload this letter as a separate file labeled 'Response to Reviewers'.- A marked-up copy of your manuscript that highlights changes made to the original version. You should upload this as a separate file labeled 'Revised Manuscript with Track Changes'.- An unmarked version of your revised paper without tracked changes. You should upload this as a separate file labeled 'Manuscript'.Please submit your revised manuscript by Jul 05 2021 11:59PM. If you will need more time than this to complete your revisions, please reply to this message or contact the journal office at plosone@plos.org. Please include the following items when submitting your revised manuscript:

We look forward to receiving your revised manuscript.

Kind regards,

Andrej M Kielbassa

Academic Editor

PLOS ONE

Journal Requirements:

2. Please include captions for ALL your Supporting Information files at the end of your manuscript, and update any in-text citations to match accordingly. Please see our Supporting Information guidelines for more information: http://journals.plos.org/plosone/s/supporting-information.

Reviewers' comments:

Reviewer's Responses to Questions

**Comments to the Author**

1. Is the manuscript technically sound, and do the data support the conclusions?

Reviewer #1: Partly

Reviewer #2: Yes

Reviewer #3: Yes

Reviewer #4: Partly

Reviewer #5: Yes

2. Has the statistical analysis been performed appropriately and rigorously? 

Reviewer #1: Yes

Reviewer #2: Yes

Reviewer #3: Yes

Reviewer #4: Yes

Reviewer #5: Yes

3. Have the authors made all data underlying the findings in their manuscript fully available?

Reviewer #1: Yes

Reviewer #2: Yes

Reviewer #3: Yes

Reviewer #4: Yes

Reviewer #5: Yes

4. Is the manuscript presented in an intelligible fashion and written in standard English?

Reviewer #1: No

Reviewer #2: Yes

Reviewer #3: Yes

Reviewer #4: Yes

Reviewer #5: Yes

5. Review Comments to the Author

Reviewer #1: The study brings new information and its objective can be relevant. Although a lot of work has been done, this submission has concerns that are listed below:

- Abstract:

Line 22: “To assess the prevalence of unculturable bacteria in dentoalveolar abscess (DAA) and related periapical lesions.” What lesions?

Line 26: Which month of 2020 was the search conducted?

Methods: Which were the eligibility criteria? What type of studies was included? Were the risk of bias, meta-regression and publication bias assessed? How was the effect measured? What type of statistical analysis was performed?

I suggest rewriting the results as follows: Approximately 13% (%95 CI: 7-23%) of the cumulative...

On the results the author affirms “Country moderator significantly ( P -value = 0.05) affects the diversity summary proportion.”, but it was not mentioned on the methods.

The conclusion is not in accordance with the objective of the study.

I suggest rewriting the conclusion as follows: “The moderate, yet significant prevalence of unculturable bacteria in such lesions, SUGGESTS that they are likely to play,...”

- Introduction

Authors should provide more information about the importance of the knowledge on the prevalence of unculturable bacteria in the dentoalveolar abscess and related periapical lesions to improve the background and also enrich the discussion of the results.

Line 51: “initiate the periapical pathology that eventually results in a DAA and its sequelae” What sequelae?

I think the authors should discuss DAA and the related periapical lesions.

Why did the author choose the Nair’s classification?

Line 60: “Periapical granuloma is a misnomer as it does not contain granulomatous tissue”. I suggest that the authors include new and current references to confirm this statement.

Line 62: Authors could include referenced to this statement.

Authors could use a single terminology to DAA, apical abscess and periapical abscess throughout the study.

I think it is necessary to clarify why it would be important to compile data on this topic.

According to PRISMA checklist “Provide an explicit statement of questions being addressed with reference to participants, interventions, comparisons, outcomes, and study design (PICOS).” Therefore, authors could include the terms “DAA and related periapical lesions” on the objective of the study (line 90). If possible, specify which “related periapical lesions”.

- Materials and Methods

The study could be rewritten and qualified according to the PRISMA Preferred reporting items for systematic reviews and meta‐analyses, available at https://onlinelibrary.wiley.com/doi/full/10.1111/iej.13118

Lines 96 and 98 – Fig 1 should be moved to the results section.

Line 105 – Authors could review the search period. In the abstract, the authors affirm that “databases were systematically searched from 1990 to 2020”. In the methodology they affirm that “the search was limited to human studies published 105 over a decade, from 1990/01/01 to 2019/12/31”. The reference number 40 corresponds to a study published in April 2020.

Have reviewers been previously calibrated to ensure inter‐rater reliability for the search and for the risk of bias assessment?

The exclusion criteria must be selected previously to conducting the study, and not according to the results. This would characterize a selection bias. The criteria presentation leaves doubt whether the criteria were previously established or not. In addition, the number of articles excluded according to each criterion must be presented in the results, not in the methods.

Line 123, 129, 131 and 132: Fig 1, Table 1, 2, 3 and S2 should be moved to the results section.

S2 Table and Table 3 are redundant.

Line 163: in this section, please include the customized checklist used to Risk of bias assessment (S3 Table).

Line 169: “Finally, all studies with a minimum score of 6/8 for 170 were included in the review.” Was it a criteria of inclusion or the result of the Risk of bias assessment?

Line 195: Please specify what is PCA in the first time you use this term on the text.

Line 201: Authors do not mention the duplicates removal. In fact, according to the flowchart, 14,819 studies were identified. 39 were duplicates and, therefore, removed.

Authors identified more than 14,000 potentially relevant articles during the electronic search, but at the end of the process only 16 articles were included.

A more refined search strategy should have been used for a more accurate result.

Why did the authors include only 13 studies in the meta-analysis? Why were 3 studies excluded?

Line 201 and 209: “dentoalveolar abscess (DAA) and related periapical lesions.” What lesions?

Line 232: S3 table should be moved to the methods section.

Lines 240 to 245: wouldn't this paragraph be a DISCUSSION of the results?

Why did the authors include acute abscess AND abscess? Why were the studies that evaluated Chronic DAA. not included in the meta-analysis? Couldn't it be a bias?

Line 274, 277, 280: The author refers to figure 2A, 2B and 2C, but this figure does not specify which is ABC.

Meta regression: Is there any limitation by using this analysis? Because there are few studies that evaluated unculturable bacteria in “periapical abscess” and “acute DAA” (type of lesion). Data would not show the real effect of the lesion type on prevalence of unculturable bacteria. Besides that, could the sample collection procedure be a specific moderator?

Line 338 to 347: Wouldn't this paragraph be a METHODOLOGY?

Line 348 to 373: For the diversity of unculturable bacteria in periapical abscess, the study number 8 “is an outlying study but not influential, so it was not removed.” But, for abundance estimation, the “tests reveal that the 3rd study [33] is an influential study on the effect of summary proportion”. Why this study was not removed?

Discussion:

Since the number of bacteria is normally greater and more virulent in symptomatic and purulent infections than in chronic and asymptomatic infections, could the inclusion of studies that do not specify what type of ADA in the meta-analysis characterize a bias in the results?

Line 417: This was the diversity found for all abscesses.

Lines 418 and 419: Authors could rewrite this sentence for better understanding.

Line 424 AND 439: Are authors suggesting any explanation for the country's interference? Is the collection method important?

More information regarding the importance of the knowledge on the prevalence of unculturable bacteria in dentoalveolar abscess and related periapical lesions should be included in this section. The authors could improve the discussion. What is the importance of knowing the frequency, abundance and diversity of these bacteria? What are the suggestions for new studies? Do the authors suggest an explanation for the significant publication bias for the frequency estimate of unculturable bacteria and for the outlier? Regarding chronic ADA, can different results be found?

What is the contribution of this study? There could be a last paragraph summarizing the findings.

Conclusions:

Authors state that the prevalence of unculturable bacteria is ABUNDANT and at the conclusion of the abstract they state that it is MODERATE. Please, review these statements.

The conclusion could be more objective and in accordance with the aim of the study.

Reviewer #2: This manuscript discusses the role unculturable bacteria in the dentoalveolar abscess and how it effects on the periapical lesions in depth through systematic review and Meta-analysis as study design. The meta analysis for this study is very good. The number of included studies in this study is expected due to new idea and the number of studies related to the topic is very small and future studies are required in this field. I think this paper eligible to publish in peer review journal like this journal.

Reviewer #3: The authors should revise the nomenclature of periapical lesions and acute apical abscess and change them in entire manuscript according to the guidelines of the AAE (diagnosis).

Dentoalveolar abscess may involve lesions of not only endodontic origin

Two studies (Jacinto et al. 2007 and Vengerfeldt et al. 2014) included in the review did not have their data clarified regarding the diagnosis. Have the authors contacted the corresponding authors to clarify the diagnosis gap? It seems that in the Jacinto et al (2007) the paper point samples were collected from root canals of symptomatic teeth (i.e. with acute apical abscess).

Great number of the studies included in this review were from Brazil. The authors highlighted a high discrepancy in the results found in these studies. Is there any hypothesis for this diversity of uncultivable bacteria found in studies carried out in the Brazilian population?

Please clarify the origin of the sample collected. For instance, Jacinto el al collected samples from the root canals of symptomatic teeth. On the other hand, Siqueira and Rocas (2007) and many other authors collected samples from the purulent collection.

Another point that needs to clarify is the presence of abscess in primary or in adult dentition. For example, Zhang et al studied the presence of abscess in primary teeth (children).

It would be interesting to explain the differences between uncultivable, unculturable and uncultivated bacteria.

Reviewer #4: Review Comments to the Author:

Please use the space provided to explain your answers to the questions above. You may also include additional comments for the author, including concerns about dual publication, research ethics, or publication ethics. (Please upload your review as an attachment if it exceeds 20,000 characters) (Limit 200 to 20000 Characters)

Could you please see my attached review, I have added all my comments to the Author in an attachment.

Reviewer #5: Abstract

- Revise for uniformity, and stick to the Journal's Guidelines. "( P -value = 0.05)" must read "(p = 0.050)". Revise thoroughly.

- With your conclusions, please stick exclusively to your aims. See: "To assess the prevalence of uncultivable bacteria in dentoalveolar abscess (DAA) and related periapical lesions." Now, you conclude that "unculturable bacteria in such lesions (...) are likely to play, a yet unknown, critical role in the pathogenesis and progression of the disease". This is something different, and you surely will agree.

- "More search remains to be done on the communal behaviour, virulence, and pathogenicity in this ecosystem." This is a common phrase only, but not an answer to your research question.

- Please stick to the word maximum allowed here, to provide complete information.

Intro

- With respect to the research question, this section is considered much too long. Please shorten significantly, and elaborate both aims and objectives more clearly.

- What about your null hypothesis?

Meths

- Why did you restrict your inclusion criteria to English?

Results

- This has been meticulously elaborated.

Disc

- Please do not use authors' names with your text (these will be acknowledged with your reference list). Instead, do focus on your main thoughts.

- This section is considered easily intelligible.

Concl

- Please see comments given above. With your Conclusions, please stick exclusively to your revised aims. Do not simply repeat your results here. Instead, provide a reasonable and generalizable extension of your outcome.

- See, for example, "Despite the remarkable diversity of uncultivable bacteria in these lesions, their role in the pathogenesis is yet to be determined. A positive correlation was shown between the abundance and frequency of individual uncultivable bacteria. Peptostreptococcus sp. oral clone CK035 showed the highest abundance and frequency. Hybridization techniques appear to be more reliable in detecting the abundance and frequency of individual unculturable bacteria in periapical lesions." All these aspects would seem right, and, thus, these thoughts might be copied & pasted to your Disc section. However, these are not considered conclusions deducible from your study.

Refs

- Full of minor and major shortcomings. Please revise for uniform formatting.

- Again, stick to the Journal's guidelines, and consult some recently published Plos One papers.

- doi numbers are missing.

6. PLOS authors have the option to publish the peer review history of their article (what does this mean?). If published, this will include your full peer review and any attached files.

Reviewer #1: No

Reviewer #2: **Yes: **Faisal Turki Alghamdi

Reviewer #3: No

Reviewer #4: No

Reviewer #5: No

---

## [Author Response · Author response to Decision Letter 0]

1 Jul 2021

30th June 2021

Editor-in-Chief

PLOS One

Dear Editor,

 Re: Manuscript entitled Prevalence of unculturable bacteria in the periapical abscess: A systematic review and meta-analysis

Thank you for the five reviewer reports on the above manuscript. We have now addressed virtually all the points raised by the reviewers as described in the point-to-point rebuttal in the attached file. Some of the reasons why we preferred the described methodology as well as the narrative edits are explained therein.

We acknowledge that the comments of the reviewers have significantly improved the manuscript and hope that it will now be acceptable for publication in PLOS One.

Thank you again.

Best regards,

Rifat Hamoudi

Point-To-Point Response

We thank the reviewers for the valuable and insightful comments. Responses to each of the comments raised are outlined below in addition to other minor changes that have been done throughout the manuscript.

Reviewer #1

- Abstract

Comment #1: Line 22: “To assess the prevalence of unculturable bacteria in dentoalveolar abscess (DAA) and related periapical lesions.” What lesions?

Response: Thank you for this comment. Taking into consideration your comment #13 in the introduction, the text was modified on lines 22 to 23.

Comment #2: Line 26: Which month of 2020 was the search conducted?

Response: The text was modified on lines 25 to 26.

More clarification for the reviewer: At first, we conducted a Boolean search for published research from 1990/01/01 to 2019/12/31. Subsequent publications, if any, up to the date of the manuscript submission was also alerted by an activated research engine, and an additional manual search which was conducted for publications between 2020/01/01 to 2020/05/31 (the period of statistical analysis and drafting the manuscript).

Comment #3: Methods: Which were the eligibility criteria? What type of studies was included? Were the risk of bias, meta-regression and publication bias assessed? How was the effect measured? What type of statistical analysis was performed?

Response: Thank you for this comment. All these points were modified and clarified further in the abstract, except for eligibility criteria. Given the limited word count of 300 words and based on previous systematic review and meta-analysis article formats used in PLOS One journal, we did not include this information in the abstract, but it is outlined extensively in the methods section. In the abstract, the text on lines 26 to 32 was modified.

Comment #4: I suggest rewriting the results as follows: Approximately 13% (%95 CI: 7-23%) of the cumulative…

Response: The text was modified on line 36.

Comment #5: On the results the author affirms “Country moderator significantly ( P -value = 0.05) affects the diversity summary proportion.”, but it was not mentioned on the methods.

Response: We thank the reviewer for this comment. According to your previous comment #3 in the abstract section, we mentioned the meta-regression and moderators on lines 27 to 29.

Comment #6: The conclusion is not in accordance with the objective of the study. I suggest rewriting the conclusion as follows: “The moderate, yet significant prevalence of unculturable bacteria in such lesions, SUGGESTS that they are likely to play,…”

Response: We thank the reviewer for this comment. According to this comment and reviewer #5/ comment #2, we have modified the conclusion on lines 46 to 49.

- Introduction

Comment #7: Authors should provide more information about the importance of the knowledge on the prevalence of unculturable bacteria in the dentoalveolar abscess and related periapical lesions to improve the background and also enrich the discussion of the results.

Response: Thank you for this comment. We have now added a paragraph to the introduction on lines 96 to 101.

Comment #8: Line 51: “initiate the periapical pathology that eventually results in a DAA and its sequelae” What sequelae?

Response: We thank the reviewer for this important point. Taking into consideration reviewer #3/comment #1. The text on lines 53 to 56 was modified.

Comment #9: I think the authors should discuss DAA and the related periapical lesions.

Response: We thank the reviewer for this comment, and we have also considered your comment #13 and reviewer #3/comment #1 about DAA term. On lines 52 to 56, the text was modified, and it briefly described the periapical abscess, radicular cyst, and periapical granuloma formation.

On lines 72 to 83, we have described the formation of periapical abscess.

We did not describe the formation of radicular cyst and periapical granuloma in any detail as they have not been included in the meta-analysis.

Comment #10: Why did the author choose the Nair’s classification?

Response: Thanks for this important question. According to this comment and reviewer #4/comment #2 we have modified the text on lines 57 to 71 and it is clearly now answering this comment.

Comment #11: Line 60: “Periapical granuloma is a misnomer as it does not contain granulomatous tissue”. I suggest that the authors include new and current references to confirm this statement.

Response: We have included another new reference Shah et al. (2017) [1] as they discussed the types of the diseases that contain granulomatous inflammation, and periapical granuloma was not included in this classification. However, according to reviewer #4/comment #2, we have deleted this sentence because we revised the classification paragraph on lines 57 to 71.

Comment #12: Line 62: Authors could include referenced to this statement.

Response: Thank you for this suggestion. We have now included an additional reference Gutmann et al (2009) [2] on line 73.

Comment #13: Authors could use a single terminology to DAA, apical abscess and periapical abscess throughout the study.

Response: We agree with the reviewer on this suggestion. In accordance with the reviewer #3/comment #1 and as most of the included studies used the term “periapical abscess” we used “periapical abscess” throughout the manuscript.

Comment #14: I think it is necessary to clarify why it would be important to compile data on this topic.

Response: Thank you for your comment. According to your comment #7, we have responded by editing the paragraph which explains the importance of unculturable bacteria in generally and specifically in the oral cavity on lines 87 to 104.

Comment #15: According to PRISMA checklist “Provide an explicit statement of questions being addressed with reference to participants, interventions, comparisons, outcomes, and study design (PICOS).” Therefore, authors could include the terms “DAA and related periapical lesions” on the objective of the study (line 90). If possible, specify which “related periapical lesions”.

Response: We thank the reviewer for this comment. As our research question did not include an intervention, we did not strictly use the PICOS research question framework and as Plos One journal’s guidelines accept other research question like CoCoPop [3]. However, we mentioned our research question on lines 111 to 118.

- Materials and Methods

Comment #16: The study could be rewritten and qualified according to the PRISMA Preferred reporting items for systematic reviews and meta‐analyses, available at https://onlinelibrary.wiley.com/doi/full/10.1111/iej.13118

Response: We thank the reviewer for this comment and accordingly modification have been performed to further follow the PRISMA guideline throughout the manuscript.

Comment #17: Lines 96 and 98 – Fig 1 should be moved to the results section.

Response: Fig 1 was moved to the results section line 210 as indicated in the manuscript, and Fig 1 legend was moved to the results section lines 228 to 229.

Comment #18: Line 105 – Authors could review the search period. In the abstract, the authors affirm that “databases were systematically searched from 1990 to 2020”. In the methodology they affirm that “the search was limited to human studies published 105 over a decade, from 1990/01/01 to 2019/12/31”. The reference number 40 corresponds to a study published in April 2020. Have reviewers been previously calibrated to ensure inter‐rater reliability for the search and for the risk of bias assessment?

Response: Thank you for this important comment. We have added a clarification in our response to your comment #2. The text on lines 129 to 131 was modified.

The time periods that have been mentioned in S1 Table were also modified to be “1990/01/01 to 2020/05/31”.

Inter‐rater reliability was done using Cohen's kappa coefficient and it was included in the method section on lines 136 to 142 and on lines 157 to 160. Then in the result section we added the results of Cohen's kappa on lines 260 to 263.

Comment #19: The exclusion criteria must be selected previously to conducting the study, and not according to the results. This would characterize a selection bias. The criteria presentation leaves doubt whether the criteria were previously established or not. In addition, the number of articles excluded according to each criterion must be presented in the results, not in the methods.

Response: Thank you for this comment and we agree with the reviewer. We did in fact decide on our exclusion criteria before conducting the research, but we found that mentioning them again caused some repetition in the text when justifying the reasons for exclusion for some studies. However, a paragraph about the exclusion criteria was added to the method section on lines 132 to 135.

The paragraph that explains the reasons and the number of the excluded articles was moved from the methods section to the results section lines 210 to 222.

Comment #20: Line 123, 129, 131 and 132: Fig 1, Table 1, 2, 3 and S2 should be moved to the results section.

Response: Thank you for this comment. Figure and tables were moved as suggested to the results section to be first mentioned as follows: (Fig 1) was moved to the result section, line 210, (Table 1) was moved to the result section line 240, (Table 2) was moved to results section line 268, (Table 3) was moved to the result section line 418, and (S2 Table) which after changing the order became (S4 Table ) and was moved to the result section line 268.

Comment #21: S2 Table and Table 3 are redundant.

Response: We thank the reviewer for this comment. Table 3 and S2 Table (which is now S4 Table after changing the order of supplementary tables) are slightly redundant, but it is essential to be kept separated and we have included one of them as supplementary data. Table 3 included information and data used for secondary outcomes, which have been analysed using PCA, and this table is a part of S4 Table, but it is essential to be kept alone as the unculturable bacteria were given symbols (from 1 to 35). These have been used to indicate the correspondent’s bacterial name, which have not been mentioned in the text and indicated the distribution of the bacteria relative to abundance, frequency, and sequence technique variables in Fig 3 and Fig 4. S4 Table included the forest plot figures for diversity, abundance, and frequency and explained how some studies were included in one forest plot but not in another forest plot due to the missing data in abundance or frequency for example.

Comment #22: Line 163: in this section, please include the customized checklist used to Risk of bias assessment (S3 Table).

Response: We thank the reviewer for this important suggestion. S3 Table becomes now S2 Table after changing the order of supplementary tables and was added to the method section/ Risk of bias assessment/ Line 157.

Comment #23: Line 169: “Finally, all studies with a minimum score of 6/8 for were included in the review.” Was it a criteria of inclusion or the result of the Risk of bias assessment?

Response: We thank the reviewer for this important comment. The mentioned sentence was moved from methods section to the results section because this was not our selection criterion; it was a subsequent criterion for quality check assessment with some modification on lines 263 to 264.

Comment #24: Line 195: Please specify what is PCA in the first time you use this term on the text.

Response: Thank you for this comment. The text on lines 200 to 201 was modified.

Comment #25: Line 201: Authors do not mention the duplicates removal. In fact, according to the flowchart, 14,819 studies were identified. 39 were duplicates and, therefore, removed.

Response: Thank you for this important comment. Information about deduplication was added to lines 206 to 209.

Comment #26: Authors identified more than 14,000 potentially relevant articles during the electronic search, but at the end of the process only 16 articles were included. A more refined search strategy should have been used for a more accurate result.

Response: This is an important point that was carefully considered, but in the current context we are providing the results that were elicited by diligent analyses by the authors. Indeed, at the beginning, we decided to include all the possibilities that allowed us to source phylotypes of unculturable bacteria. As the reviewer is aware, molecular microbiology sequencing and the study of unculturable bacteria are relatively new subject, and hence most of the older articles devoid of this term had to be rejected (i.e., 45 articles rejected as they did not state the ‘unculturable phylotypes’ despite that they used culture-independent methods for bacterial identification). In addition, as we have explained in the text the perusal of full-texts revealed approximately 60 articles that used molecular analytical methods of periapical lesions, but the vast proportion of these did not specify the histological lesion type, i.e., whether it is a radicular cyst or periapical granuloma. This led to an elimination of a majority of the filtered articles as well leading to the final result of 16 articles that were evaluated.

Comment #27: Why did the authors include only 13 studies in the meta-analysis? Why were 3 studies excluded?

Response: Thank you for this comment. For the first excluded study Siqueira and Rocas [4], we edited the paragraph to be clearer, on lines 273 to 279. For the other two studies Yun et al. [5] and Rolph et al. [6], the paragraph was edited on lines 302 to 304.

Comment #28: Line 201 and 209: “dentoalveolar abscess (DAA) and related periapical lesions.” What lesions?

Response: Thank you for this comment. We have clarified this further on lines 206 to 208.

Comment #29: Line 232: S3 table should be moved to the methods section.

Response: Thank you for this comment. S3 Table which is now S2 Table after changing the order was moved to the method section line 157. 

Comment #30: Lines 240 to 245: wouldn't this paragraph be a DISCUSSION of the results?

Response: Thank you for this suggestion. The mentioned paragraph which is now on lines 273 to 279 explains why some studies were excluded from the meta-analysis, we are not discussing the results of these studies in comparison to other studies. Furthermore, on lines 302 to 304, you can find that we explained the reasons for the exclusion of other studies Yun et al [5] and Rolph et al. [6] from meta-analysis. We believe this paragraph cannot be included in the discussion.

Comment #31: Why did the authors include acute abscess AND abscess? Why were the studies that evaluated Chronic DAA. not included in the meta-analysis? Couldn't it be a bias?

Response: Thank you for this comment. In this systematic review and meta-analysis, we included both acute and chronic forms of periapical abscess. Two studies Jacinto et al. [7] and Vengerfeldt et al. [8] did not mention in their paper whether the abscess samples were acute or chronic and therefore we termed them throughout our study simply as “abscess” and on lines 280 to 283.

Our answer for the second part of this comment (Why were the studies that evaluated Chronic DAA. not included in the meta-analysis? Couldn't it be a bias?), is that: According to our answer to your comment #27: Why did the authors include only 13 studies in the meta-analysis? Why were 3 studies excluded?), we modified the text on lines 294 to 304 to clarify why these two studies on chronic abscess were excluded from the meta-analysis.

Comment #32: Line 274, 277, 280: The author refers to figure 2A, 2B and 2C, but this figure does not specify which is ABC.

Response: We thank the reviewer for this very important comment. We have corrected this in Figure 2 as uploaded during the resubmission.

Comment #33: Meta regression: Is there any limitation by using this analysis? Because there are few studies that evaluated unculturable bacteria in “periapical abscess” and “acute DAA” (type of lesion). Data would not show the real effect of the lesion type on prevalence of unculturable bacteria. Besides that, could the sample collection procedure be a specific moderator?

Response: We thank the reviewer for this comment. We chose meta-regression for the following reasons: We looked for variables or moderators that may significantly affect our summary proportion such as lesion type, country of study origin, and the type of sequencing techniques. In addition, when determining heterogeneity, we preferred to use moderators over subgrouping, because using subgrouping for lesion types will yield a forest plot with only two studies, Vengerfeldt et al and Jacinto et al [59, 60] which is statistically inaccurate.

On the other hand, we did not investigate sampling procedure as a moderator because sequencing methods are sensitive techniques and need only a very small microbial sample to be investigated if we compared between paper point and aspiration procedure, taking into consideration an aseptic condition in both procedures had to be guaranteed. Guentsch et al. found a similar microbial load when using different sampling procedures of the gingival crevicular fluid [9]

Comment #34: Line 338 to 347: Wouldn't this paragraph be a METHODOLOGY?

Response: Thanks for this important comment. This paragraph was moved to the methods section lines 189 to 199. 

Comment #35: Line 348 to 373: For the diversity of unculturable bacteria in periapical abscess, the study number 8 “is an outlying study but not influential, so it was not removed.” But, for abundance estimation, the “tests reveal that the 3rd study [33] is an influential study on the effect of summary proportion”. Why this study was not removed?

Response: We thank the reviewer for this important comment. The reviewer is correct, the 3rd study [10] is an influential outlying study, but we decided to keep it because the summary proportion when leaving out this study revealed minor change on abundance summary proportion from 5% to 4%, while leaving out other studies like Flynn et al. [11] and Yang et al. [12] had a more prominent effect on abundance summary proportion from 5% to 7% in both studies (Flynn et al. [11] and Yang et al. [12]) although they are not influential outlying studies. For this reason, we decided to keep this study. Accordingly, it is important to investigate all these parameters to reach the better decision of exclusion. We have clarified this further on lines 395 to 398.

-Discussion:

Comment #36: Since the number of bacteria is normally greater and more virulent in symptomatic and purulent infections than in chronic and asymptomatic infections, could the inclusion of studies that do not specify what type of ADA in the meta-analysis characterize a bias in the results?

Response: We thank the reviewer for this important comment. We included the studies that did not mentioned the type of abscess whether acute or chronic in this review because in our research question we looked for periapical abscess in general. Furthermore, we have tested the heterogeneity using lesion type moderator to resolve this bias and whether it has a significant effect on diversity, abundance, and frequency of unculturable bacteria in periapical abscess.

Comment #37: Line 417: This was the diversity found for all abscesses.

Response: Yes. We measured the diversity of unculturable bacteria in periapical abscess in 13 studies as illustrated in Table 2, line 306.

Comment #38: Lines 418 and 419: Authors could rewrite this sentence for better understanding.

Response: Thank you. The text was modified on lines 472 to 474.

Comment #39: Line 424 AND 439: Are authors suggesting any explanation for the country's interference? Is the collection method important?

Response: Thank you for this important comment. We added references to explain the significance in country moderator for diversity on lines 476 to 481.

Regarding our explanation whether the collection methods is important? is that; It could be important, but when we performed the analysis, we expected that sampling procedure does not have a significant effect on our results according to a previous study [9] which found that the type of sampling procedure did not affect the bacterial load. Taking into consideration that the methods used in the included studies have used molecular methods, which are very sensitive methods for bacterial detection including unculturable bacteria.

Comment #40: More information regarding the importance of the knowledge on the prevalence of unculturable bacteria in dentoalveolar abscess and related periapical lesions should be included in this section. The authors could improve the discussion. What is the importance of knowing the frequency, abundance and diversity of these bacteria? What are the suggestions for new studies? Do the authors suggest an explanation for the significant publication bias for the frequency estimate of unculturable bacteria and for the outlier? Regarding chronic ADA, can different results be found?

Response: We thank the reviewer for this comment, the requested information was added and divided into four sections as follows:

1- More information regarding the importance of the knowledge on the prevalence of unculturable bacteria in dentoalveolar abscess and related periapical lesions should be included in this section. The authors could improve the discussion. What is the importance of knowing the frequency, abundance and diversity of these bacteria?

Response: A paragraph on the importance of knowledge on bacterial diversity in oral cavity and periapical lesions was added to the discussion section on lines 464 to 471.

A paragraph on the importance of knowledge on bacterial abundance in oral cavity and periapical lesions was added to the discussion on lines 487 to 489.

A paragraph on the importance of knowledge on bacterial frequency in oral cavity and periapical lesions was added to the discussion in lines 500 to 502.

2- What are the suggestions for new studies?

Response: We have added a paragraph as a suggestion for new studies on lines 545 to 551.

3- Do the authors suggest an explanation for the significant publication bias for the frequency estimate of unculturable bacteria and for the outlier?

Response: We thank the reviewer for this important comment. We added our explanation in the discussion section on lines 505 to 516.

4- Regarding chronic ADA, can different results be found?

Response: Generally, yes, we are expecting a different unculturable microbial profile between acute and chronic periapical abscess, but whether this difference is significant or not, this needs more research to be done on the chronic periapical abscess with larger sample size because still the real identification of unculturable bacteria is less than the culturable one as indicated in our study. Santos et al. [13] found a significant higher diversity in microbial content in acute periapical abscess in comparison to the chronic periapical abscess with the possibility that more diverse microbial content is very likely to be related to the collective pathogenicity for many endogenous diseases [14, 15]. 

Comment #41: What is the contribution of this study? There could be a last paragraph summarizing the findings.

Response: We thank the reviewer for this suggestion. We have added a last paragraph in lines 540 to 551.

- Conclusions

Comment #42: Authors state that the prevalence of unculturable bacteria is ABUNDANT and at the conclusion of the abstract they state that it is MODERATE. Please, review these statements. The conclusion could be more objective and in accordance with the aim of the study.

Response: Thank you for all the valuable comments. The text on lines 553 to 563 was modified.

Reviewer #2

Comment: This manuscript discusses the role unculturable bacteria in the dentoalveolar abscess and how it effects on the periapical lesions in depth through systematic review and Meta-analysis as study design. The meta analysis for this study is very good. The number of included studies in this study is expected due to new idea and the number of studies related to the topic is very small and future studies are required in this field. I think this paper eligible to publish in peer review journal like this journal.

Response: We thank the reviewer for their valuable opinion and positive comments.

Reviewer #3

Comment #1: The authors should revise the nomenclature of periapical lesions and acute apical abscess and change them in entire manuscript according to the guidelines of the AAE (diagnosis). Dentoalveolar abscess may involve lesions of not only endodontic origin

Response: American Association of Endodontists (AAE) guidelines (https://www.aae.org/specialty/wp-content/uploads/sites/2/2017/07/endodonticdiagnosisfall2013.pdf) were revised and all dentoalveolar abscess (DAA) terms were replaced with periapical abscess as also recommended by the reviewer #1/comment #13.

Comment #2: Two studies (Jacinto et al. 2007 and Vengerfeldt et al. 2014) included in the review did not have their data clarified regarding the diagnosis. Have the authors contacted the corresponding authors to clarify the diagnosis gap? It seems that in the Jacinto et al (2007) the paper point samples were collected from root canals of symptomatic teeth (i.e. with acute apical abscess).

Response: We thank the reviewer for this comment. For these two studies, we did not contact the authors as the necessary information for the diagnosis was obtained from their presented data but after consensus understanding and decision between the two dentists (A.M.A, and M.A.A) included in this systematic review and meta-analysis as follows: 

In the first study (Jacinto et al. 2007) [7], we are including what exactly they have written in their article and then, we explain what they mean as follows: “The selection of patients was accomplished by anamnesis, and by clinical and radiographic examination. The following features were noted for each patient: age, gender, tooth and pulpal status, nature of pain, previous pain, tenderness to percussion, pain on palpation, mobility, presence of a sinus and its origin, presence of swelling of the periodontal tissues, depth of periodontal pocket, previous antibiotic therapy and internal condition of the root canal (e.g. presence of clear, haemorrhagic or purulent exudates). Patients who had not received antibiotic therapy in the last 6 months and who had teeth with primary endodontic infections, periapical lesions and absence of periodontal diseases were selected for this study. All the cases selected showed clinical features of endodontic abscesses.” 

Then they mentioned in Results and Discussion “ The study comprised samples from teeth with primary endodontic infections, no periodontal disease, presence of periapical radiolucency, and features of endodontic abscesses.” Our explanation for Jacinto et al. patient selection criteria is as follow: 

1. They mentioned that they “noted” but they did not specify whether these features are available in all, some, or not available at all. In another words, the results of their examination were not clearly presented but they performed the required dental examination.

2. We have found that the word “noted” which means “examined” was applied to the following features {age, gender, tooth and pulpal status, nature of pain, previous pain, tenderness to percussion, pain on palpation, mobility, presence of a sinus and its origin, presence of swelling of the periodontal tissues, depth of periodontal pocket, previous antibiotic therapy and internal condition of the root canal (e.g. presence of clear, haemorrhagic or purulent exudates)}, which are all general features and are not specified characteristics for the sample or the clinical status). So, we cannot say that “we noted (age and tenderness to percussion)” and explain this sentence that there is tenderness to percussion (positive response) and apply this meaning to age and say “there is an age” this is not a clear meaning. Therefore, we decided that “noted” mean “examined or checked” but not “found, or available”. 

Then they mentioned “The study comprised samples from teeth with primary endodontic infections, no periodontal disease, presence of periapical radiolucency, and features of endodontic abscesses”. In this paragraph we took each sentence alone, e.g., samples are primary endodontic infection/ there is no periodontal disease/ there is a periapical radiolucency/ there are features of endodontic abscess. Accordingly, we understood and decided that these samples belong to periapical abscess and not periodontal abscess, but we were not sure whether it is an acute or a chronic form. 

In the second study (Vengerfeldt et al. 2014) [8], the authors were extensively and comprehensively performed the dental examination and mentioned the features of each group in Table 1 as follow: “The subject population was composed of 12 antibiotic-naive patients (ages ranging from 27–66 years) attending the Clinic of Stomatology at the University of Tartu, Estonia. Patients came to the clinic for root canal treatment or extraction between September 2010 and April 2011. Thorough anamnesis (systemic and local diseases, previous treatment, hygiene habits, allergy, and so on), intraoral status, and periapical x-rays were taken, which were all necessary for the upcoming treatment. To be included in the study, subjects had to have good systemic health. A thermal test was performed using cold and hot. A cold test was performed with Endo-Frost (-50⸰C) (Roeko, Langenau, Germany) and a cotton pellet (size 00) (Roeko); the frozen cotton pellet was held on the isolated and dried tooth on the restoration-free surface for about 2–5 seconds or until pain was felt. The hot test was performed using silicone polisher (HiLuster; Kerr Corp, Orange, CA) with a 1:1 contra-angle handpiece (W&H, B€urmoos, Austria) without air and water cooling by touching the restoration-free tooth surface with about 4000 rpm for about 5 seconds or until the patient felt pain. The electric pulp test or the vitality test was performed with the Elements Diagnostic Unit (Sybron Endo, Orange, CA) according to the manufacturer’s instructions or until pain was felt. The probe was touched to the restoration-free part of the isolated tooth until pain was felt. A percussion test was performed with a mirror handle using gentle and uniform tapping on the occlusal and horizontal side of each tooth, and sound teeth were registered as zero feeling; the same kind of tapping was performed on the accused tooth, and the feeling of the patient was compared and described. The palpation test was performed using uniform and solid pressure with the right index finger on the tip area of the root on both sides of the alveolar bone; the tooth was palpated by applying pressure on the tooth both vertically and horizontally. This was done bilaterally on both sides of the jaw to consider anatomic differences. Periapical radiographs were taken by an experienced radiologist using the Planmeca Prostile Intra X-Ray unit (Planmeca OY, Helsinki, Finland) with the RVG 6100 sensor (Carestream Dental LLC, Atlanta, GA) at a parallel angle with RINN yellow (posterior) or blue (anterior) sensor holder (Dentsply Rinn, Elgin, IL). PA radiographs were analyzed using the Trophy DICOM program (Kodak). Exclusion criteria were as follows: the presence of periodontal pockets greater than 5 mm, horizontal and vertical root fracture, deep carious lesions that made the tooth unrestorable, roots with previously apicectomized root tips with or without retrograde fillings, or any severe systemic condition like diabetes or immune suppression. In addition, subjects who received antibiotic or anti-inflammatory therapy in the previous 6 months were excluded. None of the sampled teeth presented posts, crowns, or bridges. Five of the investigated teeth were diagnosed with primary chronic apical periodontitis (pCAP) and 3 with secondary apical periodontitis (sCAP). Four teeth with a periapical abscess (evolved from apical periodontitis) were included as the controls. Clinical data are presented in Table 1.”

Accordingly, the necessary dental examination was mentioned in the text above in their study and the results of their diagnosis were presented in Table 1, so we believe that they did the data presentation of dental examination but without any discrimination of acute periapical abscess from chronic periapical abscess when they mentioned the list of bacteria derived from each sample. In their Table 2, as you can see that all periapical abscess acute or chronic are listed as periapical abscess. For that reason, we decided to include them as “periapical abscess” only during our systematic review and meta-analysis.

Comment #3: Great number of the studies included in this review were from Brazil. The authors highlighted a high discrepancy in the results found in these studies. Is there any hypothesis for this diversity of uncultivable bacteria found in studies carried out in the Brazilian population?

Response: We thank the reviewer for this comment. We have already included in the discussion our explanation for this heterogeneity in diversity scores between studies with some modification on lines 476 to 481.

Comment #4: Please clarify the origin of the sample collected. For instance, Jacinto el al collected samples from the root canals of symptomatic teeth. On the other hand, Siqueira and Rocas (2007) and many other authors collected samples from the purulent collection.

Response: We thank the reviewer for this comment. In Table 1, line 254, we mentioned the sample collection procedure including aspiration and paper point. Whenever there is an aspiration there is a purulent collection, while using paper point is related to sample collection from the root canal. However, for more clarification we added to column 9, sampling origin as indicated in the manuscript.

Comment #5: Another point that needs to clarify is the presence of abscess in primary or in adult dentition. For example, Zhang et al studied the presence of abscess in primary teeth (children).

Response: We thank the reviewer for this comment. Another column (column #7) was added to Table 1, line 254 including dentition type as indicated in the manuscript.

Comment #6: It would be interesting to explain the differences between uncultivable, unculturable and uncultivated bacteria.

Response: We thank the reviewer for this comment. This is an important point, and we agree it would be very interesting, however we believe it could be addressed in another narrative review. For this systematic review and meta-analysis, we think that this clarification will distract the reader from our goal and objectives of the study, and this should be discussed in another forum.

Reviewer #4

- Title:

Comment #1: This systematic review focused on apical abscesses. As no data on “related periapical lesions” was provided, I would suggest to remove this part of the title.

Response: We thank the reviewer for this suggestion. We have modified the title on lines 1 to 2.

- Introduction:

Comment #2: Lines 52-61: The authors used Nair (1997) as a reference for the classification of apical diseases. However, as the diagnosis in Endodontics is based on clinical signs/ symptoms and not on histopathological findings, an updated classification of apical diseases would be more appropriate (https://www.aae.org/specialty/clinical-resources/glossary-endodontic-terms/).

Response: We thank the reviewer for this comment, as this is a very important point for the updated classification of the periapical abscess. We have modified the text according to the reviewer’s recommendation to use the tenth edition of the AAE Glossary of Endodontic Terms which was updated in 2020 on lines 57 to 71.

Comment #3: Line 69: The “foregoing organisms” were not mentioned the text. Please revise this sentence.

Response: We thank the reviewer for this comment. The text was modified, and the “foregoing organisms” was replaced by the names of these organisms on lines 77 to 80.

Comment #4: Line 88: Granuloma and cysts are not sequelae of apical abscess. Please revise this sentence.

Response: We thank the reviewer for this comment. The text was modified on lines 105 to 107.

Comment #5: Lines 90-91: The secondary aim of this study was to review the possible role of non-cultivable microorganisms in the pathogenesis of apical diseases. However, the presence or abundance of bacterial DNA does not necessarily represent the bacterial activity in the disease process, as the DNA of dead bacteria can be detected by molecular methods. In addition, it has been found that bacteria with low abundance can be active in oral infections. Therefore, the secondary objective cannot be achieved due to the limitations of the studies included in this review.

Response: We thank the reviewer for the comment. The text on lines 109 to 111 was modified.

- Materials and Methods:

Comment #6: Lines 112-113: If the AAE classification of apical periodontitis (i.e., asymptomatic apical periodontitis) had been used instead of the classification based on histological findings (i.e., periapical granuloma and cyst), important information could have been extracted from the 60 articles that were excluded.

Response: In this systematic review and meta-analysis, we have focused our search strategy to include all the bacteria that have been detected in periapical abscess, radicular cyst, and periapical granuloma using molecular methods. There was a published systematic review that has been done on apical periodontitis and focused on the microbiota detected using sequencing strategy and entitled “A Systematic Review of the Root Canal Microbiota Associated with Apical Periodontitis: Lessons from Next-Generation Sequencing” [16].

Comment #7: Line 209: Please explain which periapical lesions are associated with DAA.

Response: Thank you for your comment. This paragraph was modified on lines 225 to 227.

- Outcome measures:

Comment #8: Lines 395-405: Due to the high diversity of the microbiome among individuals, most bacterial taxa were found only once. Therefore, the frequency and abundance analysis at the species level may not provide representative data.

Response: Thank you for this comment. Mentioning the taxa name of the unculturable bacteria instead of species name will be rather vague as a taxon may contain both culturable and unculturable species or phylotypes. For example, Vengerfeldt et al. (2014) [8] found in Synergistetes phylotype Pyramidobacter piscolens as culturable, while TG5 group which has a genus name only, is unculturable. Hence, we believe, it is more accurate to be specific in identifying the unculturable bacteria at their genus or species level rather than taxon level.

Comment #9: Lines 406-409: The statement that hybridization techniques are the most reliable molecular methods for detecting noncultivable bacteria should be analyzed with caution. First, it is important to note that only 5 studies were included in this analysis. Second, this finding is in contrast to the literature on microbiological methods, showing that sequencing is a reliable method for studying the oral microbiome. These issues should be further discussed.

Response: In response to the second point, our editing is on lines 334 to 339.

This mean that the “nonspecific 16S rRNA targeted gene” sequencing methods using 16S rRNA gene sequences as a general target for sequencing both the high and low abundant species wherein the high abundant species may mask the low abundant species that are usually represented by unculturable bacterial phylotypes or species [17]. On the other hand, nested PCR, restriction fragment length polymorphisms [18], and other hybridization methods [19] represented very sensitive methods for detecting even the low abundant microbial species because in these methods we are focusing on a specific and targeted area of the 16s rRNA genes that is not universal for all bacterial species but for a specific group or genus of bacteria.

Our response for the first point: We agree with the reviewer that six studies (not five studies) may not be enough for systematically concluding and affirming this result, but we think it is necessary to mention it for the first time in the context of detecting unculturable bacteria. We have modified the paragraph on lines 537 to 539.

- Conclusion:

Comment #10: Line 479: Bacterial abundance may not be correlated with their activity (please see the comments above: Lines 90-91). Please revise this sentence.

Response: We thank the reviewer for this comment. The conclusion has been modified based on your comment as well as reviewer #1/comment #42, and reviewer #5/comment #11. The conclusion now on lines 553 to 563.

Reviewer #5

- Abstract:

Comment #1: Revise for uniformity, and stick to the Journal's Guidelines. "( P -value = 0.05)" must read "(p = 0.050)". Revise thoroughly.

Response: We thank the reviewer for this important comment. We have revised the paper thoroughly and made the necessary changes to ensure uniformity. 

Comment #2: With your conclusions, please stick exclusively to your aims. See: "To assess the prevalence of uncultivable bacteria in dentoalveolar abscess (DAA) and related periapical lesions." Now, you conclude that "unculturable bacteria in such lesions (...) are likely to play, a yet unknown, critical role in the pathogenesis and progression of the disease". This is something different, and you surely will agree.

Response: We thank the reviewer for this comment and yes do in fact agree with them. We have edited the conclusion according to this comment and reviewer #1/comment #6, on lines 46 to 49.

Comment #3: More search remains to be done on the communal behaviour, virulence, and pathogenicity in this ecosystem." This is a common phrase only, but not an answer to your research question.

Response: We thank the reviewer for this suggestion. The text on lines 48 to 49 was modified.

Comment #4: Please stick to the word maximum allowed here, to provide complete information.

Response: We thank the reviewer for the comment. We have revised the word maximum whilst ensuring complete information is maintained. 

- Introduction:

Comment #5: With respect to the research question, this section is considered much too long. Please shorten significantly, and elaborate both aims and objectives more clearly.

Response: Thank you for this comment. Here we refer to another contrasting comment from a reviewer #1/comment #7 who stated, “Authors should provide more information (my emphasis) about the importance of the knowledge on the prevalence of unculturable bacteria in the dentoalveolar abscess and related periapical lesions to improve the background and also enrich the discussion of the results”. Introduction section has been revised and edited accordingly. We have edited each section according to the objective and the research question as follows:

Paragraph #1: Lines 51 to 56 explain a brief background about periapical lesions (abscess, cyst, and granuloma).

Paragraph #2: Lines 57 to 71 explain the best matched classification of periapical lesions for our study.

Paragraph #3: Lines 72 to 83 explain how the periapical abscess is formed, the bacterial role, how previously the culturable bacterial investigation has been extensively done, and the role of NGS sequencing methods in discovering the unculturable bacteria.

Paragraph #4: Lines 84 to 104 describe the universal gene sequences (16s rRNA) used in molecular methods for bacterial identification, its importance, and how it allows us to know the expected percentage of unculturable bacteria, what is the most database that have been used in this systematic review to discriminate between culturable and unculturable bacteria which is HOMD. In this paragraph, we added the importance of the unculturable bacteria and why we are studying their prevalence in periapical abscesses.

Paragraph #5: Lines 105 to 118 explain the objectives, aims, and research question of this study.

Comment #6: What about your null hypothesis?

Response: We thank the reviewer for this comment. According to PRISMA checklist and the reviewer #1/comment #15, we have included our research question in addition to some editing to the objectives on lines 105 to 118, therefore, we believe that adding the hypothesis will cause some repetition in the information. 

- Methods

Comment #7: Why did you restrict your inclusion criteria to English?

Response: We thank the reviewer for this comment. Most available data were extracted from well-recognized publications which were mostly in English. In addition, all authors are more familiar with the English language, so to ensure the consistency of data extraction and to eliminate potential biases due to translation, selecting only papers in English was seen to be sufficient for the purposes of this systematic review. 

- Results

Comment #8: This has been meticulously elaborated.

Response: We thank the reviewer and are very grateful for their positive comment.

- Discussion

Comment #9: Please do not use authors' names with your text (these will be acknowledged with your reference list). Instead, do focus on your main thoughts.

Response: We thank the reviewer for this comment. We have deleted authors names in the discussion as per the reviewer’s comment. 

Comment #10: This section is considered easily intelligible.

Response: We thank the reviewer for their positive comment. 

 - Conclusion

Comment #11: Please see comments given above. With your Conclusions, please stick exclusively to your revised aims. Do not simply repeat your results here. Instead, provide a reasonable and generalizable extension of your outcome. See, for example, "Despite the remarkable diversity of uncultivable bacteria in these lesions, their role in the pathogenesis is yet to be determined. A positive correlation was shown between the abundance and frequency of individual uncultivable bacteria. Peptostreptococcus sp. oral clone CK035 showed the highest abundance and frequency. Hybridization techniques appear to be more reliable in detecting the abundance and frequency of individual unculturable bacteria in periapical lesions." All these aspects would seem right, and, thus, these thoughts might be copied & pasted to your Disc section. However, these are not considered conclusions deducible from your study.

Response: Thank you for this comment. According to this comment, reviewer #1/comment #42, and reviewer #4/comment #10, the conclusion was modified on lines 553 to 563.

- References

Comment #12:

 - Full of minor and major shortcomings. Please revise for uniform formatting.

- Again, stick to the Journal's guidelines, and consult some recently published Plos One papers.

- doi numbers are missing.

Response: We thank the reviewer for the valuable comments. The reference list has been extensively checked and made sure to follow the journal style.

References:

1. Shah KK, Pritt BS, Alexander MP. Histopathologic review of granulomatous inflammation. Journal of Clinical Tuberculosis and Other Mycobacterial Diseases. 2017;7:1-12.

2. Gutmann JL, Baumgartner JC, Gluskin AH, Hartwell GR, Walton RE. Identify and define all diagnostic terms for periapical/periradicular health and disease states. Journal of endodontics. 2009;35(12):1658-74.

3. Jörns-Presentati A, Napp A-K, Dessauvagie AS, Stein DJ, Jonker D, Breet E, et al. The prevalence of mental health problems in sub-Saharan adolescents: A systematic review. PLoS One. 2021;16(5):e0251689.

4. Siqueira JF, Rôças IN. Molecular detection and identification of Synergistes phylotypes in primary endodontic infections. Oral diseases. 2007;13(4):398-401.

5. Yun KH, Lee H-S, Nam OH, Moon CY, Lee J-H, Choi SC. Analysis of bacterial community profiles of endodontically infected primary teeth using pyrosequencing. International journal of paediatric dentistry. 2017;27(1):56-65.

6. Rolph HJ, Lennon A, Riggio MP, Saunders WP, MacKenzie D, Coldero L, et al. Molecular identification of microorganisms from endodontic infections. Journal of clinical microbiology. 2001;39(9):3282-9.

7. Jacinto RC, Gomes BP, Desai M, Rajendram D, Shah HN. Bacterial examination of endodontic infections by clonal analysis in concert with denaturing high-performance liquid chromatography. Oral microbiology and immunology. 2007;22(6):403-10.

8. Vengerfeldt V, Špilka K, Saag M, Preem J-K, Oopkaup K, Truu J, et al. Highly diverse microbiota in dental root canals in cases of apical periodontitis (data of illumina sequencing). Journal of endodontics. 2014;40(11):1778-83.

9. Guentsch A, Kramesberger M, Sroka A, Pfister W, Potempa J, Eick S. Comparison of gingival crevicular fluid sampling methods in patients with severe chronic periodontitis. Journal of periodontology. 2011;82(7):1051-60.

10. Sakamoto M, Rocas IN, Siqueira JF, Jr., Benno Y. Molecular analysis of bacteria in asymptomatic and symptomatic endodontic infections. Oral microbiology and immunology. 2006;21(2):112-22.

11. Flynn TR, Paster BJ, Stokes LN, Susarla SM, Shanti RM. Molecular methods for diagnosis of odontogenic infections. Journal of oral and maxillofacial surgery. 2012;70(8):1854-9.

12. Yang Q-B, Fan L-N, Shi Q. Polymerase chain reaction-denaturing gradient gel electrophoresis, cloning, and sequence analysis of bacteria associated with acute periapical abscesses in children. Journal of endodontics. 2010;36(2):218-23.

13. Santos AL, Siqueira JF, Jr., Rôças IN, Jesus EC, Rosado AS, Tiedje JM. Comparing the bacterial diversity of acute and chronic dental root canal infections. PLoS One. 2011;6(11):e28088.

14. Siqueira JF, Jr., Rôças IN. Community as the unit of pathogenicity: an emerging concept as to the microbial pathogenesis of apical periodontitis. Oral surgery, oral medicine, oral pathology, oral radiology, and endodontics. 2009;107(6):870-8.

15. Kuramitsu HK, He X, Lux R, Anderson MH, Shi W. Interspecies interactions within oral microbial communities. Microbiology and molecular biology reviews. 2007;71(4):653-70.

16. Manoil D, Al-Manei K, Belibasakis GN. A systematic review of the root canal microbiota associated with apical periodontitis: Lessons from next-generation sequencing. Proteomics clinical applications. 2020;14(3):1900060.

17. McLean JS, Lombardo M-J, Badger JH, Edlund A, Novotny M, Yee-Greenbaum J, et al. Candidate phylum TM6 genome recovered from a hospital sink biofilm provides genomic insights into this uncultivated phylum. Proceedings of the national academy of sciences. 2013;110(26):E2390-E9.

18. Clusa L, Ardura A, Fernández S, Roca AA, García-Vázquez E. An extremely sensitive nested PCR-RFLP mitochondrial marker for detection and identification of salmonids in eDNA from water samples. PeerJ. 2017;5:e3045-e.

19. Podar M, Abulencia CB, Walcher M, Hutchison D, Zengler K, Garcia JA, et al. Targeted access to the genomes of low-abundance organisms in complex microbial communities. Applied and environmental microbiology. 2007;73(10):3205-14.

---

## [Decision Letter · Decision Letter 1]

19 Jul 2021

Prevalence of unculturable bacteria in the periapical abscess: A systematic review and meta-analysis

PONE-D-20-39104R1

Dear Dr. Hamoudi,

We’re pleased to inform you that your manuscript has been judged scientifically suitable for publication and will be formally accepted for publication once it meets all outstanding technical requirements.

Kind regards, congratulations and compliments, and stay healthy

Andrej M Kielbassa, Prof. Dr. med. dent. Dr. h. c.

Academic Editor

PLOS ONE

Additional Editor Comments (optional):

Reviewers' comments:

Reviewer's Responses to Questions

**Comments to the Author**

1. If the authors have adequately addressed your comments raised in a previous round of review and you feel that this manuscript is now acceptable for publication, you may indicate that here to bypass the “Comments to the Author” section, enter your conflict of interest statement in the “Confidential to Editor” section, and submit your "Accept" recommendation.

Reviewer #1: All comments have been addressed

Reviewer #2: All comments have been addressed

Reviewer #5: All comments have been addressed

2. Is the manuscript technically sound, and do the data support the conclusions?

Reviewer #1: Yes

Reviewer #2: Partly

Reviewer #5: Yes

3. Has the statistical analysis been performed appropriately and rigorously? 

Reviewer #1: Yes

Reviewer #2: Yes

Reviewer #5: Yes

4. Have the authors made all data underlying the findings in their manuscript fully available?

Reviewer #1: Yes

Reviewer #2: Yes

Reviewer #5: Yes

5. Is the manuscript presented in an intelligible fashion and written in standard English?

Reviewer #1: Yes

Reviewer #2: Yes

Reviewer #5: Yes

6. Review Comments to the Author

Reviewer #1: Dear Authors,

Thank you for accepting the recommendations.

Now, I am satisfied with the changes made and I recommend publishing the manuscript in its current format.

Reviewer #2: The authors followed the reviewer comments as well as. The paper looks good after the revision. I think it deserve published in good peer-review journal.

Reviewer #5: This revised and re-submitted draft has been thoroughly improved. Still, some minor typos would seem perfectible.

7. PLOS authors have the option to publish the peer review history of their article (what does this mean?). If published, this will include your full peer review and any attached files.

Reviewer #1: No

Reviewer #2: No

Reviewer #5: No

---

## [Editor Report · Acceptance letter]

27 Jul 2021

PONE-D-20-39104R1 

Prevalence of unculturable bacteria in the periapical abscess: A systematic review and meta-analysis 

Dear Dr. Hamoudi:

I'm pleased to inform you that your manuscript has been deemed suitable for publication in PLOS ONE. Congratulations! Your manuscript is now with our production department. 

Kind regards, 

on behalf of

Prof. Dr. med. dent. Dr. h. c. Andrej M Kielbassa 

Academic Editor

PLOS ONE